# Spatial analysis of accessibility to healthcare-related facilities in Tokyo Metropolis using geographic information systems

Tomofumi Yamazaki[1], Seiichiro Ito[1], Yoko Ino[2], Kana Sugishita[1], Koichi Kageyama[1], Mugita Sato[1], Satoshi Nakao[1,3], Kazuya Nonomura[4], Hirofumi Tamaki[4], Kazuhiro Iguchi[4], Mitsuhiro Nakamura[1]*

1 Laboratory of Drug Informatics, Gifu Pharmaceutical University, Gifu, Japan, 2 Laboratory of Pharmaceutical Health Care and Promotion, Gifu Pharmaceutical University, Gifu, Japan, 3 Department of Pharmacy, Kyushu University Hospital, Fukuoka, Japan, 4 Laboratory of Community Pharmacy, Gifu Pharmaceutical University, Gifu, Japan

* mnakamura@gifu-pu.ac.jp

## Abstract

Geographic disparities in access to health services are a growing concern in Japan as population aging and decline increase care needs and as resources concentrate in dense urban cores. Focusing on Tokyo Metropolis as a large and internally heterogeneous urban region, we used geographic information systems to evaluate spatial proximity to healthcare-related facilities—pharmacies, hospitals, clinics, dental clinics, and elderly welfare facilities—together with public transportation infrastructure. For pedestrian access, we calculated population coverage from 400 m to 3200 m Euclidean buffers around each facility. For transportation-related proximity, we calculated the proportion of facilities located within 250 m to 3000 m buffers around public transportation features (bus stops, bus routes, and railway stations). Bus stop-based facility–transportation proximity was consistently high across facility types in both the 23 special wards and the Tama region, whereas railway-station proximity displayed larger spatial variation between areas. Interpreted as an indicator for walkable proximity rather than effective service access, these results highlight where transportation connectivity and facility locations align or diverge. These findings underscore the necessity for healthcare and urban planning strategies that integrate local characteristics with transportation infrastructure.

## Introduction

The accessibility of healthcare-related facilities is an important determinant of the health outcomes of local populations and influences routine health maintenance, early disease detection, and timely treatment [1]. In Japan, healthcare resources have recently been concentrated in urban centers, whereas rural areas face limitations in both the number and functionality of facilities, leading to geographic

**Data availability statement:** All relevant data are within the paper and its Supporting Information files.

**Funding:** This research was partially supported by the Japan Society for the Promotion of Science KAKENHI grant numbers 22K10446 and 24K13369.The funders had no role in study design, data collection and analysis, decision to publish, or preparation of the manuscript.

**Competing interests:** The authors have declared that no competing interests exist.

disparities in healthcare access [2]. In Japan, national debates often emphasize urban–rural gaps in healthcare resources; however, substantial spatial inequalities can also exist within large metropolitan areas. Furthermore, with the dual demographic pressures of population aging and decline [3], healthcare needs are increasing, thereby raising a critical policy concern regarding whether residents can access the necessary medical services in a timely and appropriate manner. Addressing this challenge requires an accurate understanding of regional healthcare accessibility and formulation of resource-allocation strategies tailored to local needs.

In Japan, healthcare provision operates under a universal health insurance system in which prefectural governments are responsible for developing regional healthcare plans. Under the Medical Care Act [4], each prefecture designates secondary medical areas as planning units for coordinating hospital functions, allocating medical resources, and ensuring balanced service provision [5]. These areas were originally designed primarily for inpatient care planning; however, they also function as practical geographic units for evaluating broader healthcare accessibility, including outpatient and pharmaceutical services. Consequently, assessing accessibility within and across secondary medical areas has direct relevance to ongoing policy efforts aimed at reducing regional disparities and optimizing resource distribution. In the context of demographic aging and fiscal constraints, recent healthcare policy discussions have emphasized functional differentiation of hospitals and regional integration of care, further increasing the importance of spatial accessibility assessment at the secondary medical area level.

Healthcare accessibility is a multidimensional concept encompassing spatial accessibility, affordability, availability, and acceptability of services [6–8]. Spatial accessibility refers to the geographic relationship between populations and service locations and is commonly operationalized through measures of distance, travel time, or catchment-based coverage. In contrast, affordability concerns financial barriers to care; availability addresses the adequacy of service capacity relative to demand; and acceptability relates to sociocultural and individual preferences influencing service utilization. Among these dimensions, spatial accessibility constitutes a necessary but not sufficient condition for healthcare use. Following conceptual frameworks proposed by Shen and Tao [9] in the spatial access literature, this study focuses specifically on potential spatial accessibility as the geographic precondition for service utilization, while acknowledging that other dimensions of access fall outside the scope of the present analysis [7,8]. By isolating spatial proximity, the study seeks to clarify the geographic structure of minimum access prior to incorporating more complex determinants of healthcare utilization.

Previous domestic studies include an analysis of Tokyo Metropolis by Wang and Sadahiro [10], a study of pharmacy accessibility in Wakayama Prefecture by Kajimoto et al [11], and a 2SFCA-based study of hospital accessibility in Tochigi Prefecture by Nakamura et al. [12]. However, few studies have comprehensively evaluated and compared both walkable proximity and public-transit proximity to multiple healthcare-related facility types within Tokyo Metropolis using consistent indicators. Moreover, Wang and Sadahiro [10] focused solely on hospitals, which do not

adequately represent healthcare access as experienced in daily life. To address this limitation, the present study incorporated pharmacies, clinics, dental clinics, and elderly welfare facilities into the analytical framework.

A wide range of GIS-based approaches has been used internationally to operationalize spatial accessibility, including fixed-distance buffers [13], network-based travel-time catchments [13], gravity-type models with distance decay [14], and floating-catchment approaches such as the two-step floating catchment area (2SFCA) method [14]. We adopted a buffer-based approach to provide a transparent and reproducible baseline that can be implemented consistently across facility types and across Tokyo's planning regions using publicly available data, while acknowledging that more sophisticated network- and capacity-aware models may yield different estimates.

Internationally, GIS-based healthcare accessibility research has evolved from simple distance-based measures to more sophisticated approaches that incorporate provider-to-population ratios and distance-decay effects. Early studies conceptualized accessibility as spatial proximity between populations and facilities, often operationalized using straight-line or network buffers. Subsequently, gravity models and the 2SFCA method were developed to account for both supply capacity and population demand within defined catchments [14]. Enhanced variants further incorporated distance-decay functions and multimodal transportation systems [15]. The 2SFCA method introduced by Luo and Wang [14] has been widely applied to measure provider-to-population ratios, whereas gravity-based models address spatial interaction between supply and demand in a continuous framework. Comparative evaluations have highlighted trade-offs between cumulative opportunity measures and supply–demand models in terms of data requirements, interpretability, and policy applicability. Several international studies have also examined pharmacy accessibility as a component of primary care provision and community health resilience, particularly in urban areas characterized by heterogeneous transport infrastructure [16,17]. These methodological developments suggest that the choice of accessibility measure should be aligned with the specific analytical objective and available data environment.

Although supply–demand models provide refined estimates of potential accessibility, their implementation requires detailed information on facility capacity, service volume, and patient choice behavior, which is not consistently available at small-area levels in Japan. For policy-oriented visualization aimed at identifying areas lacking basic physical reachability, cumulative opportunity measures based on network-defined catchments remain informative and interpretable.

Accordingly, this study adopts a multimodal buffer-based approach to evaluate potential accessibility defined as physical reachability within walking, bus, and rail travel thresholds. Rather than estimating realized utilization or provider competition, the analysis focuses on first-order physical reachability as a necessary precondition for healthcare access. In this context, cumulative opportunity measures based on network-defined buffers provide a transparent and policy-relevant representation of minimum spatial access, particularly where small-area data on service volume and patient behavior are unavailable. This approach enables consistent comparison across transportation modes and facilitates clear communication with urban and health policy stakeholders. Pharmacies, in particular, serve as important points of primary care for local residents, offering daily health consultations and medication management and functioning as vital public health resources during disasters by disseminating pharmaceuticals and health-related information [18]. Therefore, evaluating accessibility to pharmacies is directly relevant to understanding the availability of healthcare services, and supporting strategies aimed at reducing disparities.

Tokyo Metropolis, located in the Kanto region near the geographic center of Japan, is a long, narrow metropolitan area extending approximately 90 km east to west and 25 km from north to south. Covering roughly 2,200 km2, Tokyo Metropolis is Japan's largest metropolitan region, with a population of approximately 14 million in 2025 [19]. Approximately 10 million residents are concentrated in the eastern section comprising 23 special wards, while approximately 4 million live in the western Tama region, which includes mountainous terrain [19]. These areas differ markedly in geography, population density, and transport infrastructure, forming a distinctive center–periphery structure within a single metropolitan jurisdiction. As Japan's largest metropolitan region, Tokyo provides a critical setting for examining how national healthcare planning frameworks operate under conditions of extreme population concentration and spatial

heterogeneity. To our knowledge, no previous study has simultaneously examined multimodal accessibility (walking, bus, and rail), multiple facility types, and secondary medical areas across the entirety of Tokyo Metropolis using small-area census data.

Accordingly, the study objectives were: (1) to quantify population coverage within 400 m to 3200 m walkable-proximity buffers around pharmacies, hospitals, clinics, dental clinics, and elderly welfare facilities; (2) to quantify facility–transportation proximity by measuring the proportion of facilities located within 250 m to 3000 m buffers of bus stops, bus routes, and railway stations; and (3) to compare these indicators across facility types, secondary medical areas, and between the 23 special wards and the Tama region.

Drawing on center–periphery spatial theory and differences in public transport network density within metropolitan regions, we hypothesize that (1) transit-based accessibility will be systematically higher in the 23 special wards; (2) walking-based coverage disparities will be more pronounced in the Tama region; and (3) pharmacies will exhibit greater intra-metropolitan variability due to market-driven location patterns in high-density commercial zones.

## Materials and methods

### Study design

This study employed a cross-sectional spatial analysis to evaluate potential healthcare accessibility across secondary medical areas in Tokyo Metropolis (Table 1, Fig 1). Facility location datasets were obtained in 2025 for pharmacies, hospitals, clinics, and dental clinics, in 2021 for elderly welfare facilities, public transportation datasets in 2022, and small-area population data from the 2020 national census (Table 2). Because these sources were produced in different years, we treat the analysis as a structural snapshot of typical facility and transportation distribution and discuss the potential impact of temporal mismatch as a limitation. The overall analytical approach consisted of: (1) constructing fixed-radius buffers around healthcare facilities and public transportation nodes; (2) estimating population coverage using areal weighting; and (3) comparing coverage indicators across medical areas and between the 23 special wards and the Tama region. The overall analytical workflow of the study is summarized in a flowchart (Fig 2). Spatial data processing, buffer construction, and overlay analyses were conducted using ArcGIS Pro (version 3.4.0; ESRI, Redlands, CA, USA). No pilot testing was conducted before full implementation of the analysis.

Table 1. Demographic and geographic attributes by secondary medical area in Tokyo Metropolis.

| | Secondary medical area | Population | Population (%) | Area (km²) | Area (%) | Population density (people/km²) | Aging rate (%) |
|---|---|---|---|---|---|---|---|
| 23 Special ward | Central ward | 946,232 | 6.8 | 63.4 | 5.0 | 14934.2 | 17.7 |
| | Southern ward | 1,171,539 | 8.4 | 83.2 | 6.6 | 14082.7 | 20.6 |
| | Southwestern ward | 1,470,746 | 10.5 | 87.4 | 7.0 | 16825.8 | 19.5 |
| | Western ward | 1,287,210 | 9.2 | 67.6 | 5.4 | 19044.4 | 19.2 |
| | Northwestern ward | 1,993,311 | 14.2 | 113.5 | 9.0 | 17557.6 | 22.0 |
| | Northeastern ward | 1,363,824 | 9.7 | 98.0 | 7.8 | 13923.7 | 24.4 |
| | Eastern ward | 1,493,241 | 10.7 | 102.3 | 8.1 | 14591.0 | 21.1 |
| Tama region | Nishitama | 379,037 | 2.7 | 57.1 | 4.5 | 6639.3 | 30.5 |
| | Minamitama | 1,437,765 | 10.3 | 322.9 | 25.7 | 4453.2 | 26.3 |
| | Western Kitatama | 657,445 | 4.7 | 89.7 | 7.1 | 7329.4 | 24.6 |
| | Southern Kitatama | 1,065,028 | 7.6 | 95.7 | 7.6 | 11132.3 | 21.3 |
| | Northern Kitatama | 747,324 | 5.3 | 76.2 | 6.1 | 9802.3 | 25.2 |

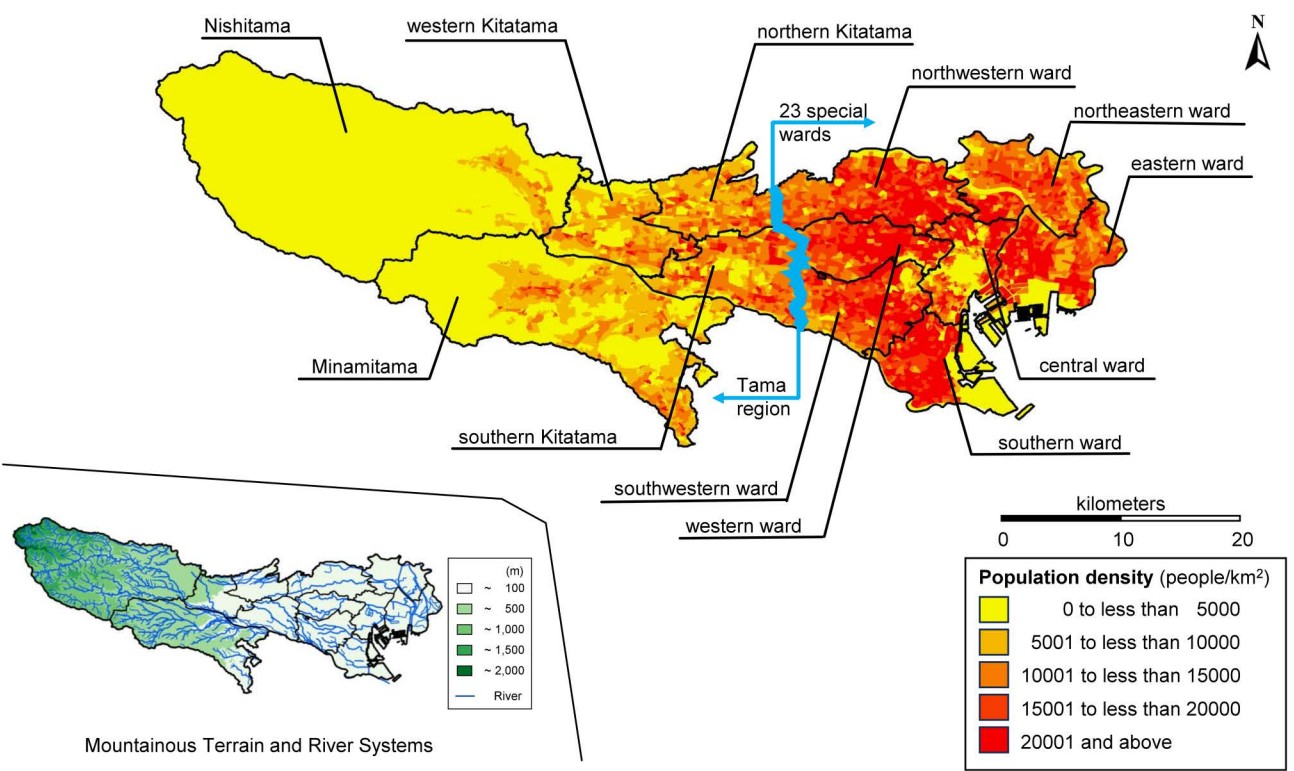

**Fig 1. Overview map of Tokyo, Japan.** Created in ArcGIS Pro using publicly available data from e-Stat, the National Land Numerical Information Download Site, and the Fundamental Geospatial Data Download Service. The source datasets are available under CC BY 4.0 or the Public Data License 1.0 (PDL 1.0), which is compatible with CC BY 4.0. The authors processed the source data and prepared the figure.

**Table 2. Data sources of boundary data.**

| Type | Data set | Source | Format | License | Reference date |
|---|---|---|---|---|---|
| Administrative boundary data | Medical area | The National Land Numerical Information download service* | JGD2011, WGS, Shapefile | CC_BY_4.0 | 2020 |
| | Small-area | e-Stat† | JGD2012, WGS, Shapefile | CC_BY_4.0 equivalent | 2020 |
| Physical environment data | Altitude | The Fundamental Geospatial Data Download Service‡ | GML | PDL 1.0 | 2016, 2025 |
| | Rivers | The National Land Numerical Information download service* | JGD2012, WGS, Shapefile | CC_BY_4.0 | 2008 |

∗ The Ministry of Land, Infrastructure, Transport and Tourism (MLIT) (https://nlftp.mlit.go.jp/ksj/).

† Statistics Bureau of Japan (https://www.e-stat.go.jp/).

‡ Geospatial Information Authority of Japan (GSI), Ministry of Land, Infrastructure, Transport and Tourism (MLIT) (https://service.gsi.go.jp/kiban/app/).

## Study region and selected healthcare facilities

In Japan, healthcare regions (iryōken) are classified into three levels according to the scale and type of healthcare provision: primary, secondary, and tertiary medical areas. Primary medical areas cover daily primary healthcare and are generally aligned with municipal boundaries. Secondary medical areas provide inpatient and specialized care and cover

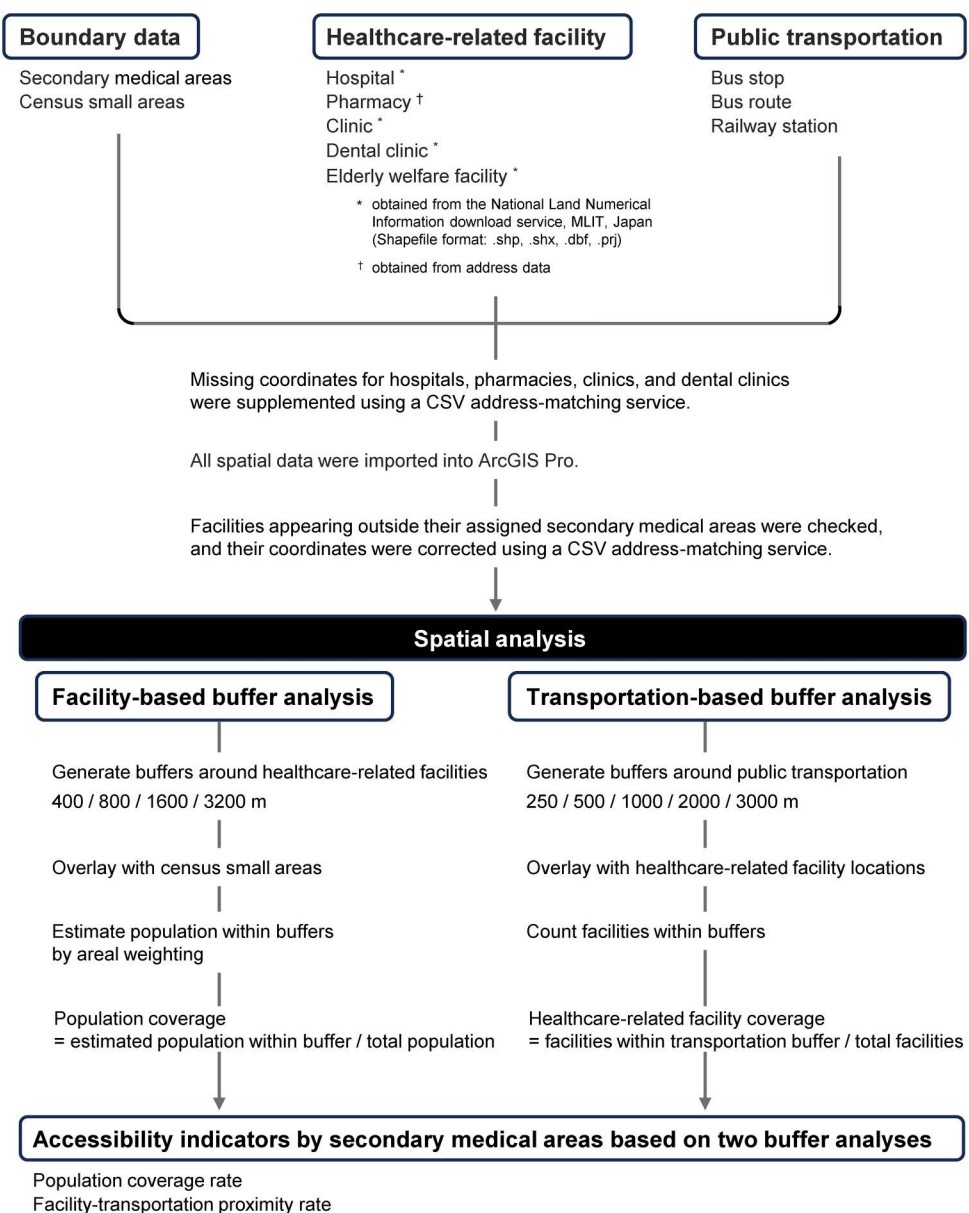

**Fig 2. Workflow for spatial accessibility analysis in Tokyo.**

multiple municipalities. Tertiary medical areas are broad-scale units offering highly specialized and advanced care, typically organized at the prefectural level.

The analysis was conducted across 12 secondary medical areas in the Tokyo Metropolis, as defined in the Tokyo Metropolitan Health and Medical Plan [20], excluding remote islands. Remote islands were excluded because facility provision and transportation connectivity are structurally different from the mainland, and small-area census polygons and public transportation layers are not directly comparable; this exclusion should be considered when interpreting citywide coverage. In the present dataset, this exclusion corresponded to 1 hospital, 18 clinics, 13 dental clinics, and 4 pharmacies, and approximately 34,891 residents (Table 1, Fig 1). These areas were central, southern, southwestern, western,

northwestern, northeastern, eastern, Nishitama, Minamitama, western Kitatama, southern Kitatama, and northern Kitatama. For descriptive summary, we report results separately for areas in the 23 special wards and areas in the Tama region. We avoid labeling these as strictly "urban" or "rural" because population density and land use vary substantially within both groups. Accordingly, this distinction is treated as an administrative and geographic classification rather than a strict urban–rural dichotomy. As shown in Table 1, there is substantial heterogeneity in population density within each group; for example, Southern Kitatama exhibits population densities comparable to central urban wards, whereas Nishitama is characterized by very low density and mountainous terrain. Therefore, in addition to this policy-relevant binary classification that reflects the institutional distinction between the 23 special wards and the Tama region, we also analyzed secondary medical areas using population density as a continuous variable. This two-level approach allows us to (1) align with the established planning framework in Tokyo Metropolis and (2) capture variation in accessibility associated with density differences across areas.

The target facilities were selected to represent major components of community-based healthcare provision in Japan, including inpatient care, outpatient care, dental services, pharmaceutical services, and long-term care support. Five facility types were included: pharmacies, hospitals, clinics, dental clinics, and elderly welfare facilities (Table 3). In Japan, hospitals and clinics are defined according to the Medical Care Act [4]. Hospitals are medical institutions with 20 or more inpatient beds, whereas clinics are institutions with fewer than 20 inpatient beds or no inpatient beds [4]. Dental clinics are also legally classified as medical institutions under the same Act and provide outpatient oral healthcare services [4]. In Japan, long-term care and welfare services for older adults are provided through various types of facilities, including special nursing homes for the elderly, residential care homes, day-care centers, and geriatric health services facilities [21]. In this study, these institutions are collectively referred to as "elderly welfare facilities." Although these facilities differ in service intensity and residential function, they were analyzed collectively because the present study focuses on spatial proximity rather than service capacity or level of care. Elderly welfare facilities include heterogeneous service models (residential and day services); therefore, we interpret their accessibility results as an aggregated indicator and discuss this heterogeneity as a limitation.

**Table 3. Data sources of healthcare-related facilities and public transportation.**

| Type | Data set | Source | Format | License | Reference date |
|---|---|---|---|---|---|
| Facility location data | Pharmacy | Open data of a medical information net* | CSV | PDL 1.0 | 01/12/2025 |
| | Hospital | Open data of a medical information net* | CSV | PDL 1.0 | 01/12/2025 |
| | Clinic | Open data of a medical information net* | CSV | PDL 1.0 | 01/12/2025 |
| | Dental clinic | Open data of a medical information net* | CSV | PDL 1.0 | 01/12/2025 |
| | Elderly welfare facility | The National Land Numerical Information download service† | JGD2011, WGS, Shapefile | CC_BY_4.0 | 2021 |
| | Bus stop | The National Land Numerical Information download service† | JGD2011, WGS, Shapefile | CC_BY_4.0 | 2022 |
| | Bus route | The National Land Numerical Information download service† | JGD2011, WGS, Shapefile | CC_BY_4.0 | 2022 |
| | Railway station | The National Land Numerical Information download service† | JGD2011, WGS, Shapefile | CC_BY_4.0 | 2022 |

* The Ministry of Health, Labor and Welfare (MHLW) (https://www.mhlw.go.jp/stf/seisakunitsuite/bunya/kenkou_iryou/iryou/newpage_43373.html).

† The Ministry of Land, Infrastructure, Transport and Tourism (MLIT) (https://nlftp.mlit.go.jp/ksj/).

## Data sources, data quality, and geocoding

Pharmacy, hospital, clinic, and dental clinic location data were obtained from the Open data of a medical information net by the Ministry of Health, Labor and Welfare (MHLW) (Table 3), which provides officially registered medical institutions under the Act on Securing Quality, Efficacy and Safety of Products Including Pharmaceuticals and Medical Devices [22] and the Medical Care Act [4]. The dataset includes geographic coordinates for each facility; therefore, these coordinates were directly used for spatial analysis. The total numbers of facilities were as follows: pharmacies (N = 6,658), hospitals (N = 591), clinics (N = 12,137), and dental clinics (N = 8,293). A small number of facilities had missing coordinate information: 24 pharmacies (0.36%), 1 hospital (0.17%), 72 clinics (0.59%), and 35 dental clinics (0.42%). For these facilities, address-based geocoding was performed using the CSV Address Matching Service of the Center for Spatial Information Science, The University of Tokyo [23]. In addition, after plotting facility locations in ArcGIS Pro, facilities that appeared outside their assigned secondary medical areas were visually checked against their registered addresses, and their coordinates were corrected using the same address-based matching approach. This correction was required for 2 hospitals, 3 clinics, and 2 pharmacies (7 facilities in total), whereas no dental clinics required such correction. All facilities were successfully assigned geographic coordinates and included in the analysis.

Data on elderly welfare facilities, bus stops, bus routes, and railway stations were obtained from the National Land Numerical Information download service provided by the Ministry of Land, Infrastructure, Transport and Tourism (MLIT) (Table 3).

No formal external positional validation study was conducted. However, the use of official source datasets and the very low proportion of medical facilities requiring coordinate supplementation suggest that the geospatial completeness of the analytical dataset was high.

The datasets used in this study were derived from different years: the 2020 Population Census, elderly welfare facility data (2021) transportation network data (2022), and medical facility data for pharmacies, hospitals, clinics, and dental clinics (2025 release). Because the objective of the study is to evaluate structural spatial patterns rather than short-term temporal changes, minor temporal mismatches were considered methodologically acceptable. Major spatial configurations of transportation infrastructure and healthcare facility distribution in Tokyo are relatively stable over short periods; therefore, the impact of data year differences on aggregate accessibility indicators is considered limited.

## Accessibility indicators and analytical methods

This study operationalized spatial accessibility as geographic proximity and evaluated it using two complementary indicators. First, the population coverage rate represents the proportion of the total population residing within a specified distance of a given facility type (i.e., potential walkable proximity to services). Second, the facility–transportation proximity rate represents the proportion of facilities located within a specified distance of public transportation features (bus stops, bus routes, or railway stations), which serves as a proxy for how well facilities are connected to transit access points.

Fixed-radius pedestrian proximity buffers of 400, 800, 1600 and 3200 m were constructed around each facility. The 400 m distance corresponds to an approximately 5-minute walk and is commonly used as a standard pedestrian proximity threshold [24,25]. The 800 m buffer provides an extended threshold that is frequently used in practice for neighborhood-scale planning and for settings with lower reliance on private cars. Given that Tokyo has the lowest household car ownership rate in Japan (0.410 cars per household vs. the national average of 1.016) [26], an extended 800 m buffer was also included to capture broader pedestrian reach. Although bicycle use is common in Tokyo, the present study focuses on pedestrian proximity to provide a conservative and uniformly applicable threshold across populations. These fixed distances do not account for individual mobility constraints and may therefore overestimate spatial accessibility for elderly or mobility-impaired populations.

For transportation-related proximity, we used 250 m to 3000 m buffers to represent pedestrian proximity to transit access points and corridors; we emphasize that walking tolerance may differ by destination type (e.g., a bus stop versus

a hospital), and we therefore interpret these buffers as standardized proxies rather than behaviorally calibrated service areas.

For the facility–transportation proximity analysis, buffers of the 250 m to 3000 m were generated around bus stops, bus route centerlines, and railway stations to allow comparability between the population coverage rate and the facility–transportation proximity rate. Bus routes were buffered from their centerline geometries with equal distances applied on both sides. Although buses operate along fixed routes with designated stops, route-based buffering was adopted to reflect the distributed boarding potential along road corridors. In contrast, railway lines were excluded because trains can only be boarded at stations; thus, only railway station buffers were constructed.

To compare accessibility indicators, coverage rates were calculated for each secondary medical area and stratified by facility type and by the 23 special wards versus the Tama region classification. Differences between the 23 special wards and the Tama region were assessed using the Mann–Whitney U test. In addition, we examined the association between population density (persons per km², derived from Table 1) and aging rate with coverage indicators using Spearman's rank correlation across the 12 secondary medical areas. This complementary analysis allowed us to treat population density as a continuous variable rather than imposing a strict categorical threshold. Because this study was based on complete enumeration of publicly available facility data and aggregated population data within the defined study area, and did not involve sampling, a sample size calculation was not applicable. Statistical analyses were conducted using RStudio (2026.01.1 + 403), with a significance level of $p < 0.05$.

Population and administrative boundary data were obtained from the 2020 Population Census at lower levels comprising small areas (cho, chome, etc.) (Table 2). This spatial scale, which is more detailed than that of municipalities, was adopted because it enables fine-grained regional comparisons. The boundary dataset was sourced from the e-Stat geographic information systems (GIS) database (Table 2). All geoprocessing was conducted in ArcGIS Pro (version 3.4.0; ESRI, Redlands, CA, USA) using standard tools including Buffer, Dissolve, Intersect, Spatial Join, and Summarize Within.

In this study, we applied areal weighting to estimate spatial indicators of healthcare accessibility. Fixed-radius buffers were constructed around relevant spatial features, including healthcare facilities and public transportation nodes or routes. For each areal unit, counts (e.g., population) were scaled by the fraction of the unit's area intersecting the buffers and then summed to obtain coverage metrics. To avoid double counting, overlapping buffers were dissolved before summarization; consequently, the total buffered share of the study area does not exceed 100%. Areal weighting was implemented as: $P\_buffer = \Sigma\_i (P\_i \times (Area\_inbuffer / Area\_i))$, where $P\_i$ is the census population of small area i (Fig 3).

For each secondary medical area, the generated buffers were overlaid with small-area population data. The population within each buffer was estimated using areal weighting, by multiplying the population of each small area by the proportion of the area within the buffer (Fig 3). The population coverage rate is defined as the ratio of the buffer population to the total population of the medical area.

For transportation-related proximity, buffers of 250 m to 3000 m were generated around bus stops, bus routes, and railway stations. Bus stops represent access points for boarding and alighting, while bus route polylines represent service corridors; route buffers were generated by buffering the route centerlines, and bus stop buffers were generated around bus stop point locations. We included both features to capture complementary aspects of bus availability (access points and service corridors). Railway proximity was represented by buffers around stations because rail service can be accessed only at station locations; therefore, railway line polylines were not buffered.

Transportation buffers were overlaid with facility locations to calculate the facility–transportation proximity rate, expressed as the proportion of facilities located within each transportation buffer (Fig 2). Comparative evaluation was conducted across facility types and secondary medical areas, and we report summary differences between the 23 special wards and the Tama region as descriptive effect sizes.

 

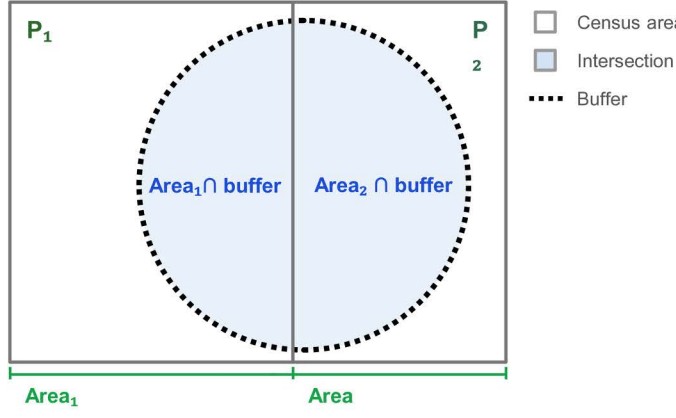

1) Concept of areal weighting

2) Model calculation example

| | $P_i$ | $Area_i$ | $Area_i \cap buf$ |
|---|---|---|---|
| Area 1 | 500 | 100 km² | 40 km² |
| Area 2 | 300 | 80 km² | 60 km² |

**Area 1**  $w_1$ = 40 / 100 = 0.40 → $P_1 \times w_1$ = 500 × 0.40 = 200
**Area 2**  $w_2$ = 60 / 80 = 0.75 → $P_2 \times w_2$ = 300 × 0.75 = 225

▼

Pbuf = $P_1 + P_2$ = 200 + 225 = 425

▼

**Estimated Buffer Population = 425 persons**

$$\mathbf{P}\textit{buf} = \Sigma i \left( \mathbf{P}i \times \frac{\mathbf{Area}i \cap \textit{\textbf{buf}}}{\mathbf{Area}i} \right)$$

**P***buf*  : Estimated population within buffer
**P***i*  : Census population of small area i
**Area***i* ∩ ***buf*** : Intersection of area i and buffer
**Area***i*  : Total area of census unit i
**Assumption**: Population is uniformly distributed within each census area.

**Fig 3. Conceptual diagram of areal weighting for population coverage estimation.**

## Ethical statement

This study used only publicly available, anonymized, aggregate data from the 2020 Population Census of Japan (via e-Stat [27]) and geospatial datasets from the National Land Numerical Information download service of the MLIT [28]. No human participants were contacted or surveyed, and no identifiable personal information was collected or analyzed. Accordingly, the study did not constitute "medical and biological research involving human subjects" as defined by Japan's Ethical Guidelines for Medical and Biological Research Involving Human Subjects [29], and institutional ethical review was not required.

## Results

Spatial buffers with radii of 400–3200 m were assigned around each pharmacy (n = 6,658), hospital (n = 591), clinic (n = 12,137), dental clinic (n = 8,293), and elderly welfare facility (n = 2,091) in Tokyo (Fig 4). Population coverage rates by secondary medical area were calculated based on these buffers (Table 4, Fig 5). Similarly, facility–transportation proximity rates were calculated using 250–3000 m buffers centered on bus stops, bus routes, and railway stations (Tables 5-9). Most facility types were concentrated in the 23 special wards, whereas the western Tama region, particularly the Nishitama area, had relatively few facilities.

For the 400 m buffer around pharmacies, hospitals, clinics, dental clinics, and elderly welfare facilities, the mean population coverage rates in the 23 special wards were 90.3%, 28.7%, 94.8%, 92.9%, and 62.1%, respectively. In the Tama region, the corresponding averages were 68.8% for pharmacies, 16.6% for hospitals, 72.8% for clinics, 68.9% for dental clinics and 41.2% for elderly welfare facilities.

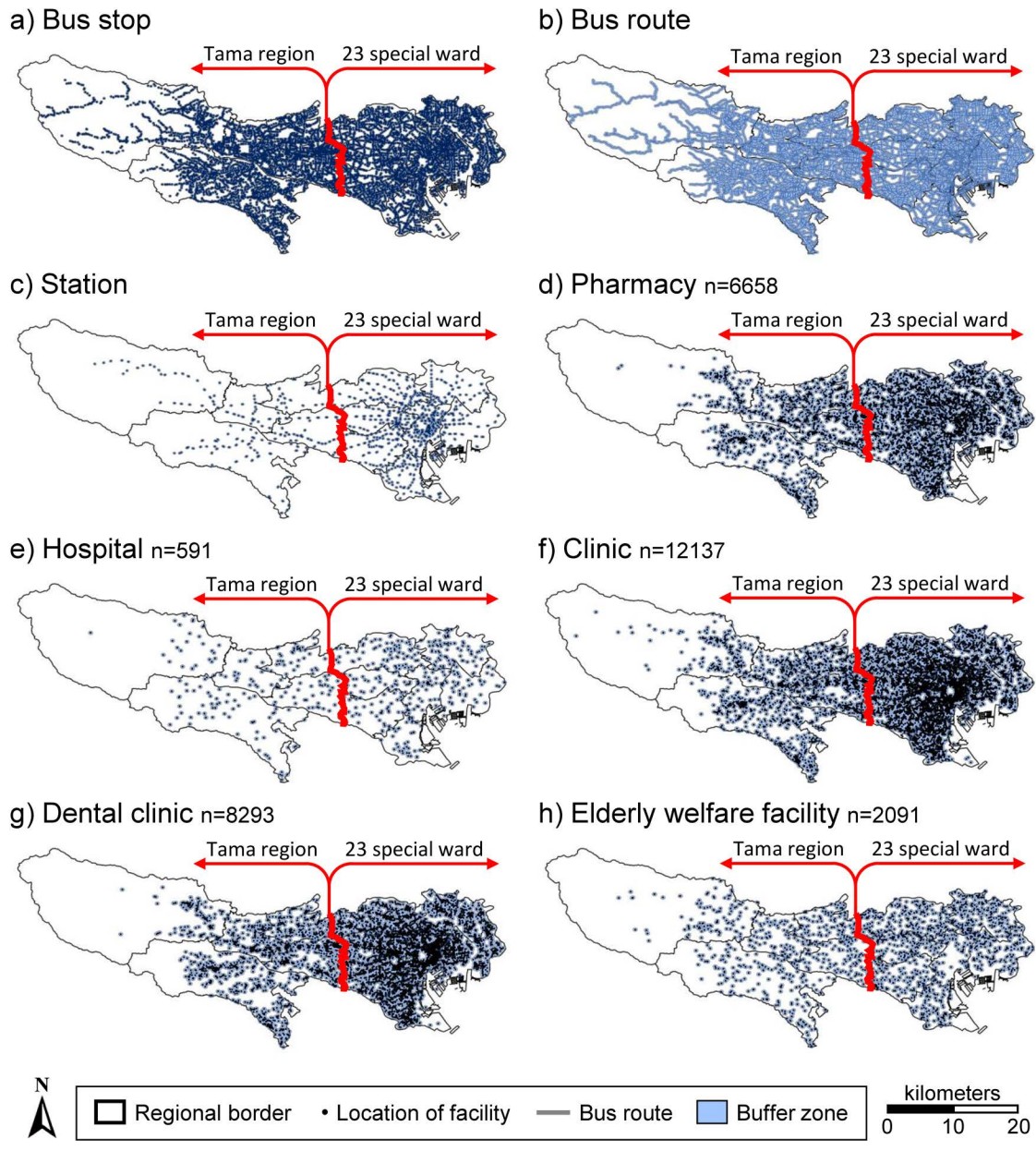

**Fig 4. Spatial distribution of healthcare-related facilities and public transportation features with 400 m facility buffers and 250 m transportation buffers.** Created in ArcGIS Pro using publicly available data from e-Stat, the National Land Numerical Information Download Site, and Open data of a medical information net. The source datasets are available under CC BY 4.0 or PDL 1.0. The authors processed the source data and prepared the figure.

When the buffer radius was extended to 800 m, population coverage rates in the 23 special wards increased to 99.7% for pharmacies, 70.3% for hospitals, 99.8% for clinics, 99.6% for dental clinics, and 95.1% for elderly welfare facilities, while coverage rates in the Tama region reached 93.3% for pharmacies, 47.2% for hospitals, 95.0% for clinics, 92.7% for dental clinics, and 83.2% for elderly welfare facilities.

**Table 4. Population coverage rates by healthcare-related facility type within specified buffer distances.**

| Secondary medical area | | Population density | 400 m buffer | | | | | 800 m buffer | | | | | 1600 m buffer | | | | | 3200 m buffer | | | | |
|---|---|---|---|---|---|---|---|---|---|---|---|---|---|---|---|---|---|---|---|---|---|---|
| | | | Pharmacy | Hospital | Clinic | Dental clinic | Elderly welfare facility | Pharmacy | Hospital | Clinic | Dental clinic | Elderly welfare facility | Pharmacy | Hospital | Clinic | Dental clinic | Elderly welfare facility | Pharmacy | Hospital | Clinic | Dental clinic | Elderly welfare facility |
| 23 Special ward | Central ward | 14934.2 | 95.4 | 29.3 | 98.9 | 97.2 | 61.2 | 100.0 | 73.9 | 100.0 | 99.8 | 92.0 | 100.0 | 97.5 | 100.0 | 100.0 | 99.4 | 100.0 | 100.0 | 100.0 | 100.0 | 100.0 |
| | Southern ward | 14082.7 | 95.2 | 27.1 | 96.9 | 96.3 | 62.2 | 100.0 | 62.5 | 99.7 | 99.9 | 93.9 | 100.0 | 97.9 | 100.0 | 100.0 | 99.2 | 100.0 | 100.0 | 100.0 | 100.0 | 100.0 |
| | South-western ward | 16825.8 | 87.6 | 21.6 | 95.8 | 93.3 | 54.4 | 99.3 | 61.1 | 99.5 | 99.6 | 94.4 | 100.0 | 94.5 | 100.0 | 100.0 | 100.0 | 100.0 | 100.0 | 100.0 | 100.0 | 100.0 |
| | Western ward | 19044.4 | 91.9 | 23.2 | 97.5 | 94.2 | 63.7 | 100.0 | 65.4 | 100.0 | 100.0 | 97.0 | 100.0 | 97.9 | 100.0 | 100.0 | 100.0 | 100.0 | 100.0 | 100.0 | 100.0 | 100.0 |
| | North-western ward | 17557.6 | 90.6 | 34.4 | 95.5 | 92.4 | 62.2 | 99.9 | 78.3 | 99.9 | 99.7 | 97.4 | 100.0 | 99.2 | 100.0 | 100.0 | 100.0 | 100.0 | 100.0 | 100.0 | 100.0 | 100.0 |
| | North-eastern ward | 13923.7 | 87.1 | 38.2 | 89.9 | 87.2 | 72.0 | 99.4 | 79.9 | 99.7 | 98.9 | 97.4 | 100.0 | 98.0 | 100.0 | 100.0 | 100.0 | 100.0 | 100.0 | 100.0 | 100.0 | 100.0 |
| | Eastern ward | 14591.0 | 84.5 | 27.2 | 89.2 | 89.9 | 59.3 | 99.0 | 70.9 | 99.7 | 99.4 | 93.6 | 100.0 | 98.2 | 100.0 | 100.0 | 99.6 | 100.0 | 100.0 | 100.0 | 100.0 | 100.0 |
| Tama region | Nishi-tama | 6639.3 | 56.6 | 11.3 | 58.0 | 51.3 | 29.5 | 81.3 | 36.9 | 86.5 | 80.2 | 69.7 | 94.2 | 82.2 | 97.6 | 92.3 | 96.7 | 98.4 | 96.0 | 99.4 | 98.3 | 99.4 |
| | Minami-tama | 4453.2 | 62.0 | 13.5 | 66.2 | 63.2 | 32.7 | 91.9 | 39.6 | 93.8 | 91.3 | 76.1 | 98.8 | 77.3 | 99.2 | 98.1 | 99.1 | 99.8 | 99.8 | 99.9 | 99.6 | 99.9 |
| | Western Kitatama | 7329.4 | 70.2 | 15.5 | 76.9 | 76.4 | 45.9 | 95.6 | 44.6 | 97.7 | 97.1 | 87.1 | 100.0 | 92.1 | 100.0 | 100.0 | 99.9 | 100.0 | 100.0 | 100.0 | 100.0 | 100.0 |
| | Southern Kitatama | 11132.3 | 79.2 | 21.8 | 85.6 | 81.9 | 46.3 | 98.3 | 59.7 | 99.2 | 98.3 | 89.9 | 100.0 | 97.1 | 100.0 | 100.0 | 100.0 | 100.0 | 100.0 | 100.0 | 100.0 | 100.0 |
| | Northern Kitatama | 9802.3 | 76.0 | 20.9 | 77.4 | 71.9 | 51.5 | 99.1 | 55.4 | 97.7 | 96.7 | 93.2 | 100.0 | 96.0 | 100.0 | 100.0 | 99.9 | 100.0 | 100.0 | 100.0 | 100.0 | 100.0 |

Unit: %.

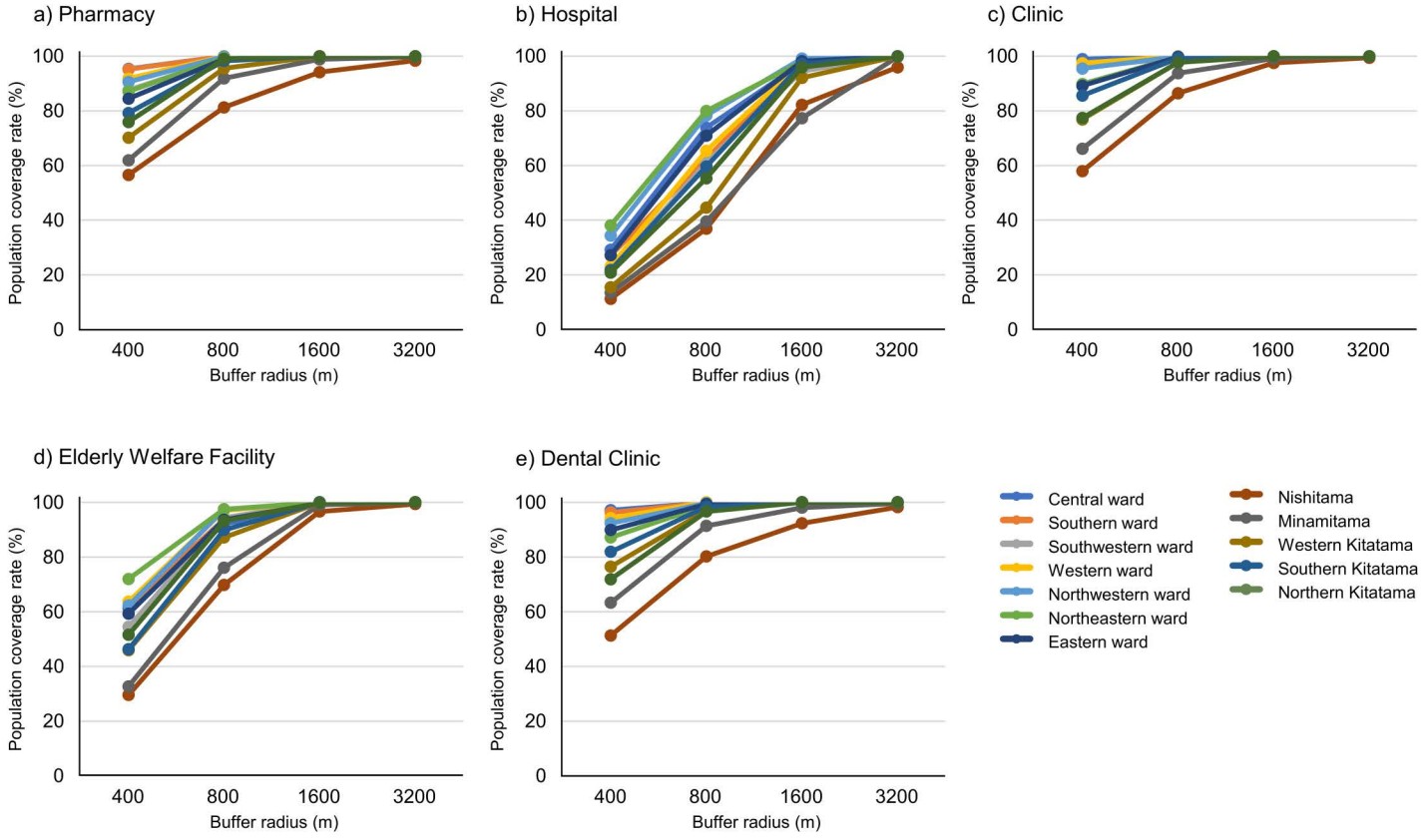

**Fig 5. Population coverage rates for medical-related facilities.**

Analysis of the facility-transportation proximity rate showed that for 250 m buffers centered on bus stops, the mean facility-transportation proximity rates in the 23 special wards were 89.2% for pharmacies, 90.2% for hospitals, 89.4% for clinics, 88.2% for dental clinics, and 81.1% for elderly welfare facilities. In the Tama region, the corresponding values were 92.1% for pharmacies, 83.3% for hospitals, 89.8% for clinics, 93.5% for dental clinics, and 80.8% for elderly welfare facilities.

Regarding buffers centered around railway stations, for the 250 m buffer, these differences were 19.4 percentage points for pharmacies, 19.8 for hospitals, 18.9 for clinics, 17.6 for dental clinics, and 8.3 for elderly welfare facilities. When the buffer radius was extended to 500 m, the differences were 23.6 percentage points for pharmacies, 32.8 for hospitals, 23.6 for clinics, 21.4 for dental clinics, and 33.0 for elderly welfare facilities. To further evaluate these descriptive patterns, non-parametric statistical analyses were performed. For population coverage indicators, the 23 special wards showed significantly higher coverage than the Tama region, particularly at shorter walking distances. At 400 m, median pharmacy coverage was 90.6% in the 23 special wards and 70.2% in the Tama region (W = 35.0, p = 0.0058), while median clinic coverage was 95.8% and 76.9%, respectively (W = 35.0, p = 0.0058). Hospital coverage at 400 m was also higher in the 23 special wards (27.2% vs. 15.5%; W = 34.0, p = 0.0094). Population density was positively correlated with pharmacy, clinic, and hospital coverage at 400 m (rho = 0.846, 0.874, and 0.678, respectively), whereas aging rate showed corresponding negative correlations (rho = −0.888 for pharmacy and −0.930 for clinic; all p < 0.05). Transportation-related proximity indicators showed the clearest differences between the 23 special wards and the Tama region. Median pharmacy proximity within 500 m of railway stations was 80.5% in the 23 special wards versus 55.7% in the Tama region (W = 35.0,

**Table 5. Facility-transportation proximity rate within 250 m buffer zone.**

| | Secondary medical area | Bus stop | | | | | Bus route | | | | | Station | | | | |
|---|---|---|---|---|---|---|---|---|---|---|---|---|---|---|---|---|
| | | Pharmacy | Hospital | Clinic | Dental clinic | Elderly welfare facility | Pharmacy | Hospital | Clinic | Dental clinic | Elderly welfare facility | Pharmacy | Hospital | Clinic | Dental clinic | Elderly welfare facility |
| 23 Special ward | Central ward | 97.8 | 100.0 | 94.7 | 95.7 | 94.4 | 100.0 | 100.0 | 98.5 | 99.2 | 7.3 | 73.4 | 50.0 | 76.9 | 75.0 | 2.6 |
| | Southern ward | 76.2 | 73.0 | 76.4 | 75.4 | 71.7 | 83.9 | 81.1 | 84.2 | 83.0 | 80.0 | 55.9 | 37.8 | 59.1 | 56.3 | 25.5 |
| | Southwestern ward | 88.0 | 97.8 | 90.0 | 86.5 | 85.9 | 91.6 | 97.8 | 92.9 | 90.8 | 90.0 | 58.3 | 39.1 | 58.3 | 58.3 | 5.3 |
| | Western ward | 92.6 | 94.7 | 92.6 | 91.4 | 73.3 | 94.4 | 94.7 | 94.4 | 94.1 | 77.8 | 63.4 | 26.3 | 63.5 | 61.4 | 17.2 |
| | Northwestern ward | 83.7 | 80.0 | 85.0 | 82.7 | 71.0 | 87.4 | 83.5 | 89.0 | 87.5 | 75.9 | 53.1 | 28.2 | 56.3 | 54.2 | 12.9 |
| | Northeastern ward | 91.8 | 89.7 | 92.6 | 91.6 | 85.9 | 96.0 | 95.4 | 95.8 | 93.9 | 92.2 | 41.9 | 20.7 | 46.0 | 42.9 | 12.6 |
| | Eastern ward | 94.5 | 96.3 | 94.2 | 93.8 | 85.6 | 97.3 | 100.0 | 97.2 | 97.4 | 91.5 | 45.6 | 24.1 | 49.2 | 42.2 | 11.4 |
| | The average of the 23 special wards | 89.2 | 90.2 | 89.4 | 88.2 | 81.1 | 92.9 | 93.2 | 93.1 | 92.3 | 73.5 | 55.9 | 32.3 | 58.5 | 55.7 | 12.5 |
| Tama region | Nishitama | 81.5 | 71.4 | 76.6 | 93.6 | 64.7 | 91.4 | 78.6 | 85.0 | 96.4 | 74.8 | 24.1 | 3.6 | 25.2 | 30.0 | 1.7 |
| | Minamitama | 95.2 | 76.1 | 91.8 | 92.4 | 77.2 | 98.3 | 83.6 | 97.6 | 97.6 | 88.6 | 35.9 | 9.0 | 39.3 | 37.5 | 4.4 |
| | Western Kitatama | 97.0 | 91.3 | 96.4 | 96.7 | 85.7 | 99.3 | 95.7 | 98.7 | 98.2 | 93.3 | 38.1 | 17.4 | 43.7 | 37.8 | 3.4 |
| | Southern Kitatama | 96.3 | 89.7 | 96.0 | 96.7 | 92.7 | 97.8 | 92.3 | 97.1 | 97.5 | 94.4 | 42.6 | 17.9 | 47.7 | 44.2 | 6.5 |
| | Northern Kitatama | 90.7 | 87.8 | 88.0 | 88.1 | 83.7 | 93.9 | 87.8 | 92.3 | 93.3 | 87.2 | 42.0 | 14.6 | 42.1 | 41.1 | 5.0 |
| | The average of the Tama region | 92.1 | 83.3 | 89.8 | 93.5 | 80.8 | 96.1 | 87.6 | 94.2 | 96.6 | 87.7 | 36.5 | 12.5 | 39.6 | 38.1 | 4.2 |
| Differences* | | −2.9 | 6.9 | −0.4 | −5.3 | 0.3 | −3.2 | 5.6 | −1.0 | −4.4 | −14.1 | 19.4 | 19.8 | 18.9 | 17.6 | 8.3 |

Unit: %.

* Value obtained by subtracting the average of the Tama region from the average of the 23 special wards.

p = 0.0058), and median clinic proximity was 82.4% versus 60.3% (W = 35.0, p = 0.0058). Station-based proximity was also strongly associated with population density, for example for pharmacy proximity within 250 m (rho = 0.888, p < 0.001) and clinic proximity within 500 m (rho = 0.881, p < 0.001), while aging rate showed strong inverse correlations with these indicators, including clinic proximity within 250 m (rho = −0.972, p < 0.001)

## Discussion

In this study, we evaluated the spatial distribution of healthcare-related facilities in Tokyo Metropolis and assessed their potential accessibility in terms of walkable proximity and Facility–transportation proximity. By integrating healthcare and transportation layers within the secondary medical-area framework, the analysis identifies where facility locations and transport connectivity align or diverge. Importantly, the indicators quantify potential access (spatial opportunity) rather than adequacy of care relative to population needs or service capacity. By integrating both healthcare and transportation perspectives, our findings highlight the necessity of considering healthcare and transportation planning as interdependent domains rather than isolated ones.

The population coverage line-graph trends further clarify the distance-dependent structure of spatial accessibility (Fig 5). For all facility types, coverage increased monotonically as buffer radius expanded; however, the rate of increase differed substantially by facility type and between the 23 special wards and the Tama region. In particular, hospitals showed relatively low coverage at shorter distances, whereas pharmacies and clinics reached high coverage more rapidly. This does

**Table 6. Facility-transportation proximity rate within 500 m buffer zone.**

| | Secondary medical area | Bus stop | | | | | Bus route | | | | | Station | | | | |
|---|---|---|---|---|---|---|---|---|---|---|---|---|---|---|---|---|
| | | Pharmacy | Hospital | Clinic | Dental clinic | Elderly welfare facility | Pharmacy | Hospital | Clinic | Dental clinic | Elderly welfare facility | Pharmacy | Hospital | Clinic | Dental clinic | Elderly welfare facility |
| 23 Special ward | Central ward | 100.0 | 100.0 | 100.0 | 100.0 | 100.0 | 100.0 | 100.0 | 100.0 | 100.0 | 100.0 | 91.7 | 89.1 | 95.4 | 95.4 | 80.6 |
| | Southern ward | 92.7 | 89.2 | 94.9 | 95.5 | 95.9 | 95.2 | 89.2 | 97.1 | 96.6 | 95.9 | 83.0 | 67.6 | 84.9 | 83.9 | 57.9 |
| | Southwestern ward | 98.5 | 100.0 | 99.1 | 98.8 | 98.8 | 99.1 | 100.0 | 99.5 | 99.4 | 99.4 | 80.5 | 69.6 | 82.4 | 82.0 | 39.4 |
| | Western ward | 99.5 | 97.4 | 99.5 | 99.8 | 97.8 | 99.7 | 97.4 | 99.5 | 99.8 | 98.3 | 82.6 | 57.9 | 86.6 | 86.3 | 53.9 |
| | Northwestern ward | 98.4 | 98.8 | 98.6 | 98.7 | 97.2 | 99.3 | 100.0 | 99.4 | 99.4 | 98.3 | 77.6 | 69.4 | 81.3 | 80.9 | 43.0 |
| | Northeastern ward | 100.0 | 100.0 | 100.0 | 99.8 | 100.0 | 100.0 | 100.0 | 100.0 | 100.0 | 100.0 | 65.3 | 42.5 | 68.8 | 65.3 | 35.6 |
| | Eastern ward | 100.0 | 100.0 | 100.0 | 99.9 | 100.0 | 100.0 | 100.0 | 100.0 | 99.9 | 100.0 | 70.8 | 61.1 | 73.7 | 67.3 | 37.3 |
| | The average of the 23 special wards | 98.4 | 97.9 | 98.9 | 98.9 | 98.5 | 99.1 | 98.1 | 99.4 | 99.3 | 98.8 | 78.8 | 65.3 | 81.9 | 80.2 | 49.7 |
| Tama region | Nishitama | 95.7 | 82.1 | 95.3 | 98.6 | 84.9 | 98.8 | 85.7 | 97.2 | 99.3 | 89.9 | 48.8 | 28.6 | 42.5 | 54.3 | 10.1 |
| | Minamitama | 99.8 | 94.0 | 99.9 | 99.8 | 99.6 | 100.0 | 97.0 | 100.0 | 100.0 | 99.6 | 50.3 | 20.9 | 57.1 | 54.8 | 17.5 |
| | Western Kitatama | 100.0 | 100.0 | 100.0 | 100.0 | 100.0 | 100.0 | 100.0 | 100.0 | 100.0 | 100.0 | 60.3 | 47.8 | 64.8 | 58.9 | 18.5 |
| | Southern Kitatama | 99.4 | 100.0 | 99.6 | 99.0 | 99.2 | 99.6 | 100.0 | 99.6 | 99.2 | 100.0 | 61.1 | 38.5 | 66.8 | 65.6 | 19.4 |
| | Northern Kitatama | 99.1 | 95.1 | 99.3 | 98.2 | 97.2 | 99.1 | 95.1 | 99.5 | 98.9 | 97.9 | 55.7 | 26.8 | 60.3 | 60.4 | 17.7 |
| | The average of the Tama region | 98.8 | 94.3 | 98.8 | 99.1 | 96.2 | 99.5 | 95.6 | 99.3 | 99.5 | 97.5 | 55.2 | 32.5 | 58.3 | 58.8 | 16.6 |
| Differences* | | −0.4 | 3.7 | 0.1 | −0.2 | 2.4 | −0.4 | 2.5 | 0.1 | −0.2 | 1.4 | 23.6 | 32.8 | 23.6 | 21.4 | 33.0 |

Unit: %.

* Value obtained by subtracting the average of the Tama region from the average of the 23 special wards.

not necessarily indicate that hospital access is spatially concentrated; rather, it likely reflects the smaller number of hospitals and their wider spacing compared with pharmacies and clinics. Furthermore, the contrast between the 23 special wards and the Tama region was most pronounced at shorter distances and became less marked as buffer size increased. This is because most indicators approached saturation at 1,600 m and 3,200 m. These findings suggest that the principal disparity lies not simply in whether facilities can be reached within broader spatial ranges, but in how quickly substantial coverage is achieved within short walking distances that are more relevant to routine healthcare access.

Bus stop-based facility-transportation proximity rate was consistently high across areas and facility types. However, under a buffer-based proximity approach, this pattern is strongly influenced by the high spatial density of bus stops and the wide geographic reach of bus corridors in Tokyo. Therefore, the results should be interpreted as indicating that many facilities are located within walkable distance of at least one bus access point, not that bus services necessarily provide superior functional accessibility (which would also depend on frequency, travel time, capacity, and connectivity). This indicates that bus networks serve as a highly comprehensive mode for accessing healthcare-related facilities irrespective of the regional context. Consequently, even in areas lacking well-developed rail systems, bus services appear to play a critical role in maintaining functional medical accessibility.

By contrast, proximity to railway stations showed larger spatial variation between the 23 special wards and the Tama region. This likely reflects the more concentrated spatial distribution of rail stations compared with bus stops, as well as the geographic constraints of peripheral and mountainous areas. While rail infrastructure can provide fast access along

**Table 7. Facility-transportation proximity rate within 1000 m buffer zone.**

| | Secondary medical area | Bus stop | | | | | Bus route | | | | | Station | | | | |
|---|---|---|---|---|---|---|---|---|---|---|---|---|---|---|---|---|
| | | Pharmacy | Hospital | Clinic | Dental clinic | Elderly welfare facility | Pharmacy | Hospital | Clinic | Dental clinic | Elderly welfare facility | Pharmacy | Hospital | Clinic | Dental clinic | Elderly welfare facility |
| 23 Special ward | Central ward | 100 | 100 | 100 | 100 | 100.0 | 100.0 | 100.0 | 100.0 | 100.0 | 100.0 | 100.0 | 100.0 | 100.0 | 99.9 | 100.0 |
| | Southern ward | 100 | 100 | 100 | 100 | 100.0 | 100.0 | 100.0 | 100.0 | 100.0 | 100.0 | 97.3 | 97.3 | 97.7 | 98.5 | 95.2 |
| | Southwestern ward | 100 | 100 | 100 | 100 | 100.0 | 100.0 | 100.0 | 100.0 | 100.0 | 100.0 | 95.8 | 95.7 | 97.0 | 96.6 | 74.1 |
| | Western ward | 100 | 100 | 100 | 100 | 100.0 | 100.0 | 100.0 | 100.0 | 100.0 | 100.0 | 98.0 | 94.7 | 98.9 | 98.0 | 94.4 |
| | Northwestern ward | 100 | 100 | 100 | 100 | 100.0 | 100.0 | 100.0 | 100.0 | 100.0 | 100.0 | 94.7 | 94.1 | 95.5 | 95.7 | 83.9 |
| | Northeastern ward | 100 | 100 | 100 | 100 | 100.0 | 100.0 | 100.0 | 100.0 | 100.0 | 100.0 | 89.9 | 74.7 | 91.3 | 89.9 | 71.5 |
| | Eastern ward | 100 | 100 | 100 | 100 | 100.0 | 100.0 | 100.0 | 100.0 | 100.0 | 100.0 | 88.6 | 79.6 | 90.4 | 89.2 | 73.6 |
| | The average of the 23 special wards | 100 | 100 | 100 | 100 | 100 | 100 | 100 | 100 | 100 | 100 | 94.9 | 90.9 | 95.8 | 95.4 | 84.7 |
| Tama region | Nishitama | 100 | 92.9 | 98.6 | 100 | 99.2 | 100.0 | 92.9 | 98.6 | 100.0 | 99.2 | 76.5 | 35.7 | 68.7 | 77.9 | 35.3 |
| | Minamitama | 100 | 100 | 100 | 100 | 100.0 | 100.0 | 100.0 | 100.0 | 100.0 | 100.0 | 70.5 | 46.3 | 74.5 | 72.5 | 40.8 |
| | Western Kitatama | 100 | 100 | 100 | 100 | 100.0 | 100.0 | 100.0 | 100.0 | 100.0 | 100.0 | 80.8 | 73.9 | 88.6 | 81.6 | 73.1 |
| | Southern Kitatama | 100 | 100 | 100 | 100 | 100.0 | 100.0 | 100.0 | 100.0 | 100.0 | 100.0 | 79.9 | 74.4 | 83.7 | 83.3 | 54.8 |
| | Northern Kitatama | 100 | 100 | 100 | 100 | 100.0 | 100.0 | 100.0 | 100.0 | 100.0 | 100.0 | 77.4 | 53.7 | 82.5 | 79.3 | 51.8 |
| | The average of the Tama region | 100.0 | 98.6 | 99.7 | 100.0 | 99.8 | 100.0 | 98.6 | 99.7 | 100.0 | 99.8 | 77.0 | 56.8 | 79.6 | 78.9 | 51.2 |
| Differences* | | 0.0 | 1.4 | 0.3 | 0.0 | 0.2 | 0.0 | 1.4 | 0.3 | 0.0 | 0.2 | 17.9 | 34.1 | 16.2 | 16.5 | 33.5 |

Unit: %.

* Value obtained by subtracting the average of the Tama region from the average of the 23 special wards.

major corridors, station-based proximity alone cannot capture network connectivity, service frequency, or first/last-mile barriers; thus, our findings should be interpreted as a baseline spatial pattern rather than a definitive evaluation of rail service performance. This spatial pattern suggests that the expansion of railway networks alone may be insufficient to address the healthcare accessibility deficits in peripheral areas.

The supplementary statistical analyses further supported these spatial patterns. Population coverage indicators were generally higher in the 23 special wards than in the Tama region, particularly at shorter walking distances, and several of these indicators were positively correlated with population density while showing negative correlations with aging rate. A similar tendency was observed for facility–transportation proximity indicators, especially those based on railway stations, where the 23 special wards consistently showed more favorable proximity patterns than the Tama region. Taken together, these findings suggest that spatial accessibility in Tokyo is shaped not only by the broad contrast between the 23 special wards and the Tama region, but also by a continuous gradient of urban density and demographic structure. In other words, areas with higher population density and lower levels of aging tend to have more advantageous configurations of healthcare-related facilities and transport access, whereas lower-density and more rapidly aging areas may face comparatively greater spatial constraints.

Particularly in the Tama region, the establishment of new hospitals or railway infrastructure is often prevented by financial and human resource limitations [30,31]. Instead, strategies such as optimizing existing bus routes, deploying community buses to support elderly mobility, integrating healthcare delivery with public transport planning, and implementing mobile

**Table 8. Facility-transportation proximity rate within 2000 m buffer zone.**

| | Secondary medical area | Bus stop | | | | | Bus route | | | | | Station | | | | |
|---|---|---|---|---|---|---|---|---|---|---|---|---|---|---|---|---|
| | | Pharmacy | Hospital | Clinic | Dental clinic | Elderly welfare facility | Pharmacy | Hospital | Clinic | Dental clinic | Elderly welfare facility | Pharmacy | Hospital | Clinic | Dental clinic | Elderly welfare facility |
| 23 Special ward | Central ward | 100.0 | 100.0 | 100.0 | 100.0 | 100.0 | 100.0 | 100.0 | 100.0 | 100.0 | 100.0 | 100.0 | 100.0 | 100.0 | 100.0 | 100.0 |
| | Southern ward | 100.0 | 100.0 | 100.0 | 100.0 | 100.0 | 100.0 | 100.0 | 100.0 | 100.0 | 100.0 | 100.0 | 100.0 | 100.0 | 100.0 | 100.0 |
| | Southwestern ward | 100.0 | 100.0 | 100.0 | 100.0 | 100.0 | 100.0 | 100.0 | 100.0 | 100.0 | 100.0 | 100.0 | 100.0 | 99.9 | 100.0 | 98.2 |
| | Western ward | 100.0 | 100.0 | 100.0 | 100.0 | 100.0 | 100.0 | 100.0 | 100.0 | 100.0 | 100.0 | 100.0 | 100.0 | 100.0 | 100.0 | 100.0 |
| | Northwestern ward | 100.0 | 100.0 | 100.0 | 100.0 | 100.0 | 100.0 | 100.0 | 100.0 | 100.0 | 100.0 | 98.4 | 97.6 | 99.0 | 99.1 | 96.5 |
| | Northeastern ward | 100.0 | 100.0 | 100.0 | 100.0 | 100.0 | 100.0 | 100.0 | 100.0 | 100.0 | 100.0 | 99.1 | 97.7 | 98.5 | 99.3 | 95.9 |
| | Eastern ward | 100.0 | 100.0 | 100.0 | 100.0 | 100.0 | 100.0 | 100.0 | 100.0 | 100.0 | 100.0 | 99.8 | 100.0 | 99.8 | 100.0 | 99.5 |
| | The average of the 23 special wards | 100.0 | 100.0 | 100.0 | 100.0 | 100.0 | 100.0 | 100.0 | 100.0 | 100.0 | 100.0 | 99.6 | 99.3 | 99.6 | 99.8 | 98.6 |
| Tama region | Nishitama | 100.0 | 100.0 | 100.0 | 100.0 | 100.0 | 100.0 | 100.0 | 100.0 | 100.0 | 100.0 | 95.7 | 64.3 | 89.7 | 95.7 | 73.1 |
| | Minamitama | 100.0 | 100.0 | 100.0 | 100.0 | 100.0 | 100.0 | 100.0 | 100.0 | 100.0 | 100.0 | 88.1 | 76.1 | 88.3 | 90.3 | 70.2 |
| | Western Kitatama | 100.0 | 100.0 | 100.0 | 100.0 | 100.0 | 100.0 | 100.0 | 100.0 | 100.0 | 100.0 | 96.4 | 100.0 | 98.2 | 97.6 | 93.3 |
| | Southern Kitatama | 100.0 | 100.0 | 100.0 | 100.0 | 100.0 | 100.0 | 100.0 | 100.0 | 100.0 | 100.0 | 96.1 | 94.9 | 98.7 | 98.5 | 91.9 |
| | Northern Kitatama | 100.0 | 100.0 | 100.0 | 100.0 | 100.0 | 100.0 | 100.0 | 100.0 | 100.0 | 100.0 | 97.1 | 95.1 | 97.6 | 98.2 | 96.5 |
| | The average of the Tama region | 100.0 | 100.0 | 100.0 | 100.0 | 100.0 | 100.0 | 100.0 | 100.0 | 100.0 | 100.0 | 94.7 | 86.1 | 94.5 | 96.1 | 85.0 |
| Differences[*] | | 0.0 | 0.0 | 0.0 | 0.0 | 0.0 | 0.0 | 0.0 | 0.0 | 0.0 | 0.0 | 4.9 | 13.3 | 5.1 | 3.7 | 13.6 |

Unit: %.

* Value obtained by subtracting the average of the Tama region from the average of the 23 special wards.

clinics or pharmacies may offer more effective and efficient solutions. This underscores the importance of resource-allocation strategies that align healthcare infrastructure with the transportation network rather than simply increasing the number of facilities or transport nodes in isolation. This study assessed spatial accessibility to medical facilities in Tokyo and sought to generate implications for urban planning. However, it did not include a comprehensive examination of Tokyo's urban planning framework, policy-making processes, or ongoing planning debates. Consequently, the extent to which the findings can be directly situated within Tokyo's current urban planning context is limited and should be interpreted with caution.

## Study limitations

Several limitations should be considered when interpreting the current study's findings.

First, spatial proximity was assessed using straight-line (Euclidean) buffers rather than actual travel paths, i.e., road networks, topography, and pedestrian infrastructure such as crossings, were not accounted for. While straight-line buffers are simpler, highly reproducible, and easily transferable to other regions—advantages over more complex methods such as the two-step floating catchment area (2SFCA) approach used by Wang and Sadahiro [10]—they can lead to underestimation of travel times in cases where a point lies within the Euclidean buffer but falls outside the catchment area when measured by road distance, as noted by Boscoe et al [32]. We therefore interpret our results as standardized proxy measures of spatial accessibility and discuss network-based alternatives (e.g., 2SFCA and travel-time catchments) as future work. These standardized distances may not reflect heterogeneous mobility constraints (e.g., older adults or

**Table 9. Facility-transportation proximity rate within 3000 m buffer zone.**

| | Secondary medical area | Bus stop | | | | | Bus route | | | | | Station | | | | |
|---|---|---|---|---|---|---|---|---|---|---|---|---|---|---|---|---|
| | | Pharmacy | Hospital | Clinic | Dental clinic | Elderly welfare facility | Pharmacy | Hospital | Clinic | Dental clinic | Elderly welfare facility | Pharmacy | Hospital | Clinic | Dental clinic | Elderly welfare facility |
| 23 Special ward | Central ward | 100.0 | 100.0 | 100.0 | 100.0 | 100.0 | 100.0 | 100.0 | 100.0 | 100.0 | 100.0 | 100.0 | 100.0 | 100.0 | 100.0 | 100.0 |
| | Southern ward | 100.0 | 100.0 | 100.0 | 100.0 | 100.0 | 100.0 | 100.0 | 100.0 | 100.0 | 100.0 | 100.0 | 100.0 | 100.0 | 100.0 | 100.0 |
| | Southwestern ward | 100.0 | 100.0 | 100.0 | 100.0 | 100.0 | 100.0 | 100.0 | 100.0 | 100.0 | 100.0 | 100.0 | 100.0 | 100.0 | 100.0 | 100.0 |
| | Western ward | 100.0 | 100.0 | 100.0 | 100.0 | 100.0 | 100.0 | 100.0 | 100.0 | 100.0 | 100.0 | 100.0 | 100.0 | 100.0 | 100.0 | 100.0 |
| | Northwestern ward | 100.0 | 100.0 | 100.0 | 100.0 | 100.0 | 100.0 | 100.0 | 100.0 | 100.0 | 100.0 | 100.0 | 100.0 | 99.9 | 100.0 | 100.0 |
| | Northeastern ward | 100.0 | 100.0 | 100.0 | 100.0 | 100.0 | 100.0 | 100.0 | 100.0 | 100.0 | 100.0 | 100.0 | 100.0 | 99.9 | 100.0 | 99.6 |
| | Eastern ward | 100.0 | 100.0 | 100.0 | 100.0 | 100.0 | 100.0 | 100.0 | 100.0 | 100.0 | 100.0 | 100.0 | 100.0 | 100.0 | 100.0 | 100.0 |
| | The average of the 23 special wards | 100.0 | 100.0 | 100.0 | 100.0 | 100.0 | 100.0 | 100.0 | 100.0 | 100.0 | 100.0 | 100.0 | 100.0 | 100.0 | 100.0 | 99.9 |
| Tama region | Nishitama | 100.0 | 100.0 | 100.0 | 100.0 | 100.0 | 100.0 | 100.0 | 100.0 | 100.0 | 100.0 | 99.4 | 78.6 | 94.9 | 75.7 | 89.1 |
| | Minamitama | 100.0 | 100.0 | 100.0 | 100.0 | 100.0 | 100.0 | 100.0 | 100.0 | 100.0 | 100.0 | 93.4 | 86.6 | 94.5 | 96.3 | 84.2 |
| | Western Kitatama | 100.0 | 100.0 | 100.0 | 100.0 | 100.0 | 100.0 | 100.0 | 100.0 | 100.0 | 100.0 | 99.7 | 100.0 | 99.8 | 99.7 | 100.0 |
| | Southern Kitatama | 100.0 | 100.0 | 100.0 | 100.0 | 100.0 | 100.0 | 100.0 | 100.0 | 100.0 | 100.0 | 100.0 | 100.0 | 100.0 | 100.0 | 100.0 |
| | Northern Kitatama | 100.0 | 100.0 | 100.0 | 100.0 | 100.0 | 100.0 | 100.0 | 100.0 | 100.0 | 100.0 | 99.7 | 100.0 | 99.8 | 99.6 | 100.0 |
| | The average of the Tama region | 100.0 | 100.0 | 100.0 | 100.0 | 100.0 | 100.0 | 100.0 | 100.0 | 100.0 | 100.0 | 98.4 | 93.0 | 97.8 | 94.3 | 94.7 |
| Differences* | | 0.0 | 0.0 | 0.0 | 0.0 | 0.0 | 0.0 | 0.0 | 0.0 | 0.0 | 0.0 | 1.6 | 7.0 | 2.2 | 5.7 | 5.3 |

Unit: %.

* Value obtained by subtracting the average of the Tama region from the average of the 23 special wards.

mobility-impaired individuals), and they do not incorporate cycling, which is common in Tokyo; thus, they should not be interpreted as behaviorally calibrated walking service areas or as indicators of emergency-care access.

Second, our analysis was limited to quantitative facility counts and did not account for functional or capacity-related attributes, such as available medical specialties, number of beds, or hours of operation. As such, the results do not reflect the actual service provision capacity. Moreover, this study included elderly welfare facilities—such as long-term care and day-care centers—in the analysis. However, their accessibility profiles differ substantially from those of other healthcare-related facilities. Residents of long-term care facilities generally have little need for external access, whereas day-care centers typically provide transportation services for users [33]. Therefore, findings related to elderly welfare facilities should be interpreted with caution.

Third, the proximity indicators used were confined to spatial proximity and did not incorporate non-geographic factors such as waiting times, medical costs, or socioeconomic determinants influencing healthcare-seeking behaviors. In addition, the areal-weighting approach assumes that population is uniformly distributed within each census polygon; this assumption may introduce error, especially in heterogeneous land-use areas.

Finally, our public-transportation layers represent infrastructure and published routes/stops, but do not capture temporal variability such as service frequency, timetable changes, or seasonal adjustments; therefore, the results reflect spatial proximity to access points rather than time-dependent service availability.

## Future research directions

Future studies should incorporate network-based analyses using road data to estimate travel times for multiple transport modes including walking, buses, and private vehicles. Evaluating spatial accessibility by the type of medical service or specialty, applying weightings based on mobility constraints among specific populations (e.g., older adults), and integrating public transport service quality indicators such as frequency and operating hours, are also important steps. Additionally, empirical assessments of the effectiveness of supplemental interventions such as mobility support programs and mobile pharmacies would provide valuable evidence for integrated healthcare and transportation policy development. Future work could also explore network Voronoi-based catchments and other network-partitioning approaches to better represent realistic service areas in dense urban networks.

## Conclusion

This study evaluated spatial proximity to healthcare-related facilities (pharmacies, clinics, hospitals, and elderly welfare facilities) in Tokyo Metropolis using GIS-based analyses that incorporated pedestrian buffers and facility-transportation proximity buffers. The results showed substantial spatial variation in proximity to railway stations between the 23 special wards and the Tama region, while proximity to bus access points was broadly high under the buffer-based metric. Interpreted as standardized proxy measures for potential access (not as measures of service capacity or utilization), these findings provide practical evidence to support regional healthcare planning and transportation policy integration within the existing secondary medical-area framework. The approach can serve as a reproducible baseline for future work that incorporates network travel times, service quality, and facility capacity.

## Supporting information

**S1 File.** e-stat terms of use (Japanese).
(PDF)

**S2 File.** e-stat terms of use (English).
(DOCX)

**S3 File. National land numerical information download site terms of use (Japanese).**
(PDF)

**S4 File. National land numerical information download site terms of use (English).**
(DOTM)

**S5 File. Digital agency public data license (version 1.0) (Japanese).**
(PDF)

**S6 File. Digital agency public data license (version 1.0) (English).**
(DOCX)

**S7 Table. Population coverage rates by healthcare-related facility type within specified buffer distances.**
(XLSX)

**S8 Table. Facility-transportation proximity rate within 250 m buffer zone.**
(XLSX)

**S9 Table. Facility-transportation proximity rate within 500 m buffer zone.**
(XLSX)

**S10 Table. Facility-transportation proximity rate within 1000 m buffer zone.**
(XLSX)

**S11 Table. Facility-transportation proximity rate within 2000 m buffer zone.**
(XLSX)

**S12 Table. Facility-transportation proximity rate within 3000 m buffer zone.**
(XLSX)

**S13 Script.** R script for Mann–Whitney U analysis of population coverage rates.
(R)

**S14 Script.** R script for Spearman correlation analysis of population coverage rates.
(R)

**S15 Script.** R script for Mann–Whitney U analysis of facility–transportation proximity rates.
(R)

**S16 Script. R script for Spearman correlation analysis of facility–transportation proximity rates.**
(R)

**S17 Table. Input data for Mann–Whitney U analysis of population coverage rates.**
(CSV)

**S18 Table. Input data for Spearman correlation analysis of population coverage rates.**
(CSV)

**S19 Table. Input data for Mann–Whitney U analysis of facility–transportation proximity rates.**
(CSV)

**S20 Table. Input data for Spearman correlation analysis of facility–transportation proximity rates.**
(CSV)

**S21 Table. Results of Mann–Whitney U analysis of population coverage rates.**
(CSV)

**S22 Table. Results of Spearman correlation analysis of population coverage rates.**
(CSV)

**S23 Table. Results of Mann–Whitney U analysis of facility–transportation proximity rates.**
(CSV)

**S24 Table. Results of Spearman correlation analysis of facility–transportation proximity rates.**
(CSV)

## Author contributions

**Conceptualization:** Tomofumi Yamazaki, Seiichiro Ito, Hirofumi Tamaki, Kazuhiro Iguchi, Mitsuhiro Nakamura.

**Data curation:** Tomofumi Yamazaki, Seiichiro Ito, Kana Sugishita, Koichi Kageyama, Mugita Sato, Satoshi Nakao, Mitsuhiro Nakamura.

**Formal analysis:** Tomofumi Yamazaki, Seiichiro Ito, Hirofumi Tamaki, Mitsuhiro Nakamura.

**Funding acquisition:** Hirofumi Tamaki, Kazuhiro Iguchi.

**Investigation:** Tomofumi Yamazaki, Seiichiro Ito, Satoshi Nakao, Kazuhiro Iguchi, Mitsuhiro Nakamura.

**Methodology:** Tomofumi Yamazaki, Seiichiro Ito, Kazuya Nonomura, Hirofumi Tamaki, Kazuhiro Iguchi, Mitsuhiro Nakamura.

**Resources:** Tomofumi Yamazaki, Mitsuhiro Nakamura.

**Software:** Mitsuhiro Nakamura.

**Supervision:** Yoko Ino, Kazuya Nonomura, Hirofumi Tamaki, Kazuhiro Iguchi, Mitsuhiro Nakamura.

**Validation:** Tomofumi Yamazaki, Seiichiro Ito, Yoko Ino, Kana Sugishita, Koichi Kageyama, Mugita Sato, Satoshi Nakao, Kazuhiro Iguchi.

**Visualization:** Tomofumi Yamazaki, Seiichiro Ito, Mitsuhiro Nakamura.

**Writing – original draft:** Tomofumi Yamazaki, Seiichiro Ito, Yoko Ino, Kazuhiro Iguchi, Mitsuhiro Nakamura.

**Writing – review & editing:** Tomofumi Yamazaki, Mitsuhiro Nakamura.

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
