## [Decision Letter · Decision Letter 0]

3 Feb 2026

PONE-D-25-58835Spatial Analysis of Accessibility to Healthcare-Related Facilities in Tokyo Metropolis Using Geographic Information SystemsPLOS One

Dear Dr. Nakamura,

Thank you for submitting your manuscript to PLOS ONE. After careful consideration, we feel that it has merit but does not fully meet PLOS ONE’s publication criteria as it currently stands. Therefore, we invite you to submit a revised version of the manuscript that addresses the points raised during the review process.

We look forward to receiving your revised manuscript.

Kind regards,

Lingye Yao, Ph.D.

Academic Editor

PLOS One

Journal Requirements:

3. We note that Figure(s) 1 and 3 in your submission contain [map/satellite] images which may be copyrighted. All PLOS content is published under the Creative Commons Attribution License (CC BY 4.0), which means that the manuscript, images, and Supporting Information files will be freely available online, and any third party is permitted to access, download, copy, distribute, and use these materials in any way, even commercially, with proper attribution. For these reasons, we cannot publish previously copyrighted maps or satellite images created using proprietary data, such as Google software (Google Maps, Street View, and Earth). For more information, see our copyright guidelines: http://journals.plos.org/plosone/s/licenses-and-copyright.

a. You may seek permission from the original copyright holder of Figure(s) 1 and 3 to publish the content specifically under the CC BY 4.0 license.

Reviewers' comments:

Reviewer's Responses to Questions

**Comments to the Author**

1. Is the manuscript technically sound, and do the data support the conclusions?

Reviewer #1: Partly

Reviewer #2: Partly

2. Has the statistical analysis been performed appropriately and rigorously? 

Reviewer #1: N/A

Reviewer #2: No

3. Have the authors made all data underlying the findings in their manuscript fully available?

Reviewer #1: Yes

Reviewer #2: Yes

4. Is the manuscript presented in an intelligible fashion and written in standard English?

Reviewer #1: Yes

Reviewer #2: Yes

5. Review Comments to the Author

Reviewer #1: Manuscript ID: PONE-D-25-58835

Title: Spatial Analysis of Accessibility to Healthcare-Related Facilities in Tokyo Metropolis Using Geographic Information Systems

INTRODUCTION SECTION REVIEW (Lines 37-70)

1. Literature Review Insufficiency

- Only two previous Japanese studies are cited (Wang and Sadahiro; Kajimoto et al.)

- No international literature on healthcare accessibility using GIS

- No discussion of methodological approaches (2SFCA, gravity models, etc.) used elsewhere

- Missing citations on pharmacy accessibility specifically

RECOMMENDATION: Expand literature review to include international GIS-based healthcare accessibility studies and explain why the simpler buffer method was chosen over more sophisticated approaches.

2. Research Questions/Objectives Unclear

- The introduction ends with a general aim "to generate evidence to facilitate..." (line 68) but lacks specific research questions

- No hypotheses stated

- Unclear what specific comparisons or relationships will be tested

3. Conceptual Framework Missing

- No discussion of what "accessibility" means conceptually

- The introduction jumps from problem to methods without theoretical grounding

- Reference [10] (Shen & Tao) on spatial access is cited later but not introduced here

RECOMMENDATION: Add a brief paragraph distinguishing spatial accessibility from other dimensions (affordability, acceptability, availability) and justify the focus on spatial proximity.

4. Policy Context Underdeveloped

- Line 44-45 mentions "critical policy concern" but doesn't elaborate

- No mention of existing Japanese healthcare policies or planning frameworks

- The later reference to secondary healthcare zones (line 75) appears without context

RECOMMENDATION: Briefly introduce Japan's healthcare system structure and relevant policies driving interest in accessibility analysis.

2. MATERIALS AND METHODS (Lines 73-155)

1. STUDY DESIGN SECTION MISSING (Lines 73-74)

The methods jump directly into "Study region and selected healthcare facilities" without describing the overall study design.

RECOMMENDATION: Add a brief study design paragraph covering:

• Study type (cross-sectional spatial analysis)

• Timeframe

• Overall analytical approach

• Software and tools used (ArcGIS Pro is mentioned line 126 but should be introduced earlier)

2. FACILITY SELECTION RATIONALE INCOMPLETE (Lines 88-101)

• Line 88: "The target facilities were of four types..." but no justification WHY these four

• Why exclude dental clinics after extracting them (line 99)?

• No operational definitions: What constitutes a "clinic" vs. "hospital"? Are these based on bed count, services, or legal designation?

• Elderly welfare facilities are defined collectively (lines 89-92) but the heterogeneity of these facilities is problematic for accessibility analysis

RECOMMENDATION:

• Provide clear inclusion/exclusion criteria for each facility type

• Define facilities according to Japanese healthcare classifications

• Justify the exclusion of dental clinics

• Consider analysing elderly welfare facilities separately or excluding them

3. DATA QUALITY AND VALIDATION (Lines 93-101)

• No discussion of data completeness or accuracy

• Geocoding quality not assessed-what was the match rate from CSV Address Matching Service?

• Different data vintages (2020 census, 2022 transportation, 2024 pharmacy) not acknowledged

• No mention of how address errors or missing data were handled

RECOMMENDATION: Add a data quality subsection addressing:

• Geocoding match rates and accuracy

• Handling of unmatched addresses

• Impact of using different data years

• Validation methods (if any)

4. ACCESSIBILITY INDICATORS (Lines 108-127)

• Lines 112-114: The "population coverage rate" is well-defined, but the "facility coverage rate" definition is potentially confusing. It's the proportion of facilities near transportation, not transportation near facilities.

• Line 115: "Walking catchment buffers"-the term "catchment" typically refers to service areas of facilities, but here it seems to mean the area around facilities accessible by walking. Clarify terminology.

• Lines 116-120: The justification for 800m extension is based on low car ownership, but:

- What about bicycle use (very common in Tokyo)?

- What about elderly or mobility-impaired populations who cannot walk 800m?

- Is 800m (about 10 minutes) reasonable for emergency care?

• Line 121: "Equivalent 400 and 800m buffers were created around the public transportation nodes"-equivalent in what sense? The meaning differs: walking to a facility vs. walking to transportation to reach a facility.

• No justification for why the same distances (400m, 800m) are appropriate for both facility access and transportation access. Walking to a bus stop vs. walking to a hospital have different tolerance thresholds.

RECOMMENDATIONS:

• Clarify conceptual distinction between facility catchment areas and transportation service areas

• Consider different buffer distances for transportation nodes (perhaps 250m and 500m as commonly used for bus stops)

• Acknowledge limitations for mobility-impaired populations

• Discuss why cycling is not considered despite being common in Tokyo

• Lines 144-152: The justification for including bus routes but excluding railway lines (lines 148-150) makes sense but is incompletely explained. Bus routes are included because passengers can board anywhere, but:

- Do the bus route data indicate where buses actually stop?

- How were bus routes buffered (centerline? both sides of road?)

- Trains can only board at stations, but railway proximity might still matter for perceived accessibility

• Lines 153-155: "Both the population and facility coverage rates were evaluated comparatively",vague. Comparative how? Between what groups? This should specify:

- Urban vs. rural comparisons

- Comparisons across facility types

- Statistical methods (none mentioned!)

RECOMMENDATIONS:

• Add explicit statistical analysis section

• Describe buffer generation for bus routes more precisely

• Acknowledge and discuss the uniform population distribution assumption

• Specify exactly what comparisons will be made and how

6. SECONDARY HEALTHCARE ZONES CLASSIFICATION (Lines 75-80, Table 1)

• Line 77: "excluding remote islands"-how many facilities/population are excluded? This could be important for understanding coverage.

• Lines 78-80: The classification into "urban" (23 special wards) vs. "rural" (Tama region) is overly simplistic:

- Table 1 shows wide variation within each category

- Southern Kitatama has 11,048.8 people/km² (quite urban)

- Nishitama has only 661.9 people/km² (truly rural)

- This binary classification may obscure important nuances

RECOMMENDATION: Consider a three-tier classification (urban/suburban/rural) based on population density thresholds, or analyze as a continuous variable.

7. MISSING METHODOLOGICAL ELEMENTS

• No sample size justification: While this is a complete enumeration, should discuss statistical power for detecting differences

• No ethical statement: Line "N/A" in ethics section—should briefly justify why ethics approval was not needed (public aggregate data)

• No definition of temporal scope: Data from 2020-2024 but no discussion of appropriate reference period

• No handling of seasonal variations: Bus routes and schedules vary seasonally in Tokyo

• No consideration of operational hours: 24-hour pharmacies vs. those closing at 6 PM have very different practical accessibility

• No pilot testing mentioned: Was the methodology tested on a subset before full implementation?

TABLE 1 (Lines 82-83)

• Column header "Population (n)" suggests a sample, but this is total population, use "Population" or "Total Population"

• Consider adding % of Tokyo total for each zone

• Consider adding age distribution (% elderly) given the focus on healthcare needs

• Area calculation precision seems excessive (63.6 km² is clear enough without decimal)

TABLE 2 (Lines 102-106)

• "Type" column includes both administrative boundaries and facility locations—consider splitting into two tables

• "Creation date" and "Acquisition date" distinction unclear-is "creation" when data were produced and "acquisition" when authors obtained them?

• Some acquisition dates in 2024, census data from 2020-should acknowledge this lag

• Footnote symbols (*, †, ‡) not consistently formatted

• URLs should be provided in full or archived (link rot is common)

FIGURE 1 (Lines 86-87)

Caption: "Created with ArcGIS Pro software", unnecessary detail for figure caption

• No scale bar visible

• Administrative boundaries not clearly distinguished

• Consider adding labels for major geographic features (rivers, mountains) mentioned in text

FIGURE 2 (Line 146)

• "Flowchart" may be misleading-this is more of a conceptual diagram showing spatial overlay operations

• Should specify which analysis (population coverage or facility coverage) is being illustrated

• The areal weighting calculation could be shown more explicitly (formula?)

• Consider adding a small example with actual numbers

2. SOFTWARE AND COMPUTATIONAL DETAILS

ArcGIS Pro version mentioned (line 126) but no details on:

• Specific tools used (Buffer, Spatial Join, etc.)

5. DISCUSSIONS

The Discussions section should be extended and more comparation with similar studies should be mentioned.

Reviewer #2: This paper examines geographic disparities in healthcare accessibility using GIS-based spatial analysis, focusing on population coverage around healthcare facilities in the Tokyo metropolitan area. The topic is relevant and timely, as understanding spatial inequalities in healthcare access is an important issue for public health and urban planning. The manuscript is generally well-structured, and the authors acknowledge several methodological limitations, which is appreciated.

However, to improve the study's overall quality, robustness, and interpretability, the authors could consider the following points.

1. The abstract and introduction motivate the study by referring to rural–urban disparities in healthcare access. However, the empirical analysis focuses exclusively on the Tokyo metropolitan area, a highly urbanised context. The authors should clarify why Tokyo alone is sufficient to address the stated motivation, or revise the framing to better align with an intra-metropolitan accessibility analysis rather than a rural–urban comparison.

2. The analysis treats all hospitals as equivalent in terms of service provision. This assumption is problematic, as hospitals differ significantly in size, service scope, and clinical capacity. While line 128 shows that Japan has 3 levels of healthcare regions, the result only shows 1 type of hospital. At a minimum, the authors should more clearly justify this simplification and explicitly limit interpretations of results to spatial proximity rather than to effective healthcare access.

3. Accessibility is measured using fixed distance buffers, which may be misleading in a dense and complex urban road network such as Tokyo. Network-based methods are well established in the GIS literature and have been widely applied to healthcare accessibility studies. The authors should justify the choice of Euclidean buffers more carefully, or at least discuss how network-based approaches might alter the results. Furthermore, several publications have implemented the Network Voronoi diagram to identify the catchment area

4. The study appears to focus largely on calculating population counts within 400 m and 800 m buffers around healthcare facilities and transport nodes. While this approach is simple, the methodological steps would benefit from clearer explanation and visualisation. Given that this is a GIS-focused study, additional schematic figures illustrating the analytical workflow and assumptions would greatly improve clarity.

5. Several figures show expected patterns, such as higher population density and higher facility coverage in central urban areas compared to peripheral areas. While these results are not incorrect, the manuscript would benefit from deeper interpretation. For example, it is unclear whether the observed accessibility levels are adequate relative to population needs, or how they compare across demographic groups.

6. The authors acknowledge important limitations, including the use of Euclidean buffers, the lack of facility capacity differentiation, and the exclusion of non-spatial factors such as waiting time or socioeconomic conditions. However, the conclusions still make relatively strong claims about healthcare accessibility. The authors should ensure that the conclusions are more tightly aligned with what the simplified methodology can realistically support.

7. In the conclusion, the author highlighted an imbalance in accessibility between the railway and the bus, where the bus has more comprehensive coverage. Under the current buffer-based approach, the finding that bus networks provide more comprehensive coverage than rail is effectively predetermined by the spatial density of bus stops. Without accounting for service frequency, capacity, or network connectivity, this conclusion risks overstating the functional accessibility contribution of bus systems.

6. PLOS authors have the option to publish the peer review history of their article (what does this mean?). If published, this will include your full peer review and any attached files.

Reviewer #1: No

Reviewer #2: No

---

## [Author Response · Author response to Decision Letter 1]

17 Apr 2026

Response to Reviewer 1

We wish to express our sincere gratitude for your thorough and meticulous review of our manuscript. Your expert assessment and well-considered remarks have been instrumental in enhancing the scholarly rigor and overall quality of our work. Each of your comments has been examined with the utmost care, and corresponding revisions have been incorporated into the manuscript where appropriate.

The following presents a detailed, point-by-point response, with page and line numbers provided to indicate the precise locations of the modifications in the revised version.

1. INTRODUCTION SECTION REVIEW (Lines 37-70)

1-1. Literature Review Insufficiency

1-1-1. Only two previous Japanese studies are cited (Wang and Sadahiro; Kajimoto et al.)

Thank you for your suggestion. In addition to the existing domestic studies by Wang and Sadahiro [10] and Kajimoto et al. [11], we have added Nakamura et al. [12]. Nakamura et al. evaluated hospital accessibility in Japan using a 2SFCA-based method, and we believe that including this study allows the present research to be positioned more appropriately within the context of spatial accessibility studies in Tokyo that focus on pharmacies and multiple types of healthcare facilities. This information has been added to the Introduction section of the revised manuscript (page 4, lines 75–77; page 5, lines 84–90).

1-1-2. No international literature on healthcare accessibility using GIS

Thank you for your suggestion. To strengthen the international context of GIS-based healthcare accessibility research, we added Owen et al. [13], Luo and Wang [14], and Luo and Qi [15]. This enabled us to reposition the present study within the major streams of research, including distance-based, network-based, gravity-model, and floating-catchment approaches. This information has been added to the Introduction section of the revised manuscript (page 5, line 84 to page 6, line 106).

1-1-3. No discussion of methodological approaches (2SFCA, gravity models, etc.) used elsewhere

Thank you for your suggestion. As major methods used in previous studies, we explicitly added the conceptual framework proposed by Guagliardo [1] and Shen and Tao [9], as well as the time-distance and fixed-distance approaches of Owen et al. [13], the 2SFCA method of Luo and Wang [14], and the E2SFCA method of Luo and Qi [15]. This allowed us to explain, within the methodological background, the relationship among gravity models, floating-catchment approaches, network-based travel-time approaches, and the fixed-distance buffer method adopted in the present study. This information has been added to the Introduction section of the revised manuscript (page 5, line 84 to page 6, line 106).

1-1-4. Missing citations on pharmacy accessibility specifically

As references on pharmacy accessibility, we added the Wakayama Prefecture study by Kajimoto et al. [11], Azuma [17] on the role of community pharmacies, the systematic review by Jagadeesan et al. [16], and Aruru et al. [18]. These additions allowed us to more clearly demonstrate the significance of including pharmacies in the present study from the perspectives of both everyday access to healthcare and response during disasters and emergencies. This information has been added to the Introduction section of the revised manuscript (page 4, lines 75–77; page 6, lines 102–104; page 7, lines 119–122).

1-1-5. RECOMMENDATION: Expand literature review to include international GIS-based healthcare accessibility studies and explain why the simpler buffer method was chosen over more sophisticated approaches.

In accordance with your suggestion, we addressed the points raised in items 1-1-1 through 1-1-5 above.

2. Research Questions/Objectives Unclear

2-1 The introduction ends with a general aim "to generate evidence to facilitate..." (line 68) but lacks specific research questions

We agree that the description of the study objectives in the original manuscript was too abstract. In the revised manuscript, we replaced the general statement of purpose with more specific study objectives that clearly indicate the focus of the analysis. This information has been added to the Introduction section of the revised manuscript (page 8, lines 138–143).

2-2 No hypotheses stated

Thank you for your suggestion. In the revised manuscript, we clarified the study objectives and explicitly stated the hypotheses based on differences in the center–periphery structure of Tokyo and the density of the public transportation network. Specifically, we proposed three hypotheses: (1) transit-based accessibility will be systematically higher in the 23 special wards; (2) walking-based coverage disparities will be more pronounced in the Tama region; and (3) pharmacies will exhibit greater intra-metropolitan variability due to market-driven location patterns in high-density commercial zones.

This information has been added to the Introduction section of the revised manuscript (page 8, lines 144–148).

2-3 Unclear what specific comparisons or relationships will be tested

In the revised manuscript, we revised the Introduction and Methods sections to specify the comparison targets more clearly. Specifically, we stated that the analysis compared facility types, buffer distances, secondary medical service areas, and the 23 special wards versus the Tama area, and that the associations with population density and aging rate were also evaluated statistically. This information has been added to the revised manuscript (page 3, line 50 to page 4, line 74; page 12, lines 175–180; page 16, line 220 to page 19, line 291; page 20, line 312 to page 21, line 323).

3. Conceptual Framework Missing

By clarifying points 1 and 2 above, we were able to clarify the conceptual framework of the study.

3-1. No discussion of what "accessibility" means conceptually

In the Introduction of the revised manuscript, we added a conceptual paragraph that frames healthcare accessibility as a multidimensional concept encompassing availability, affordability, and acceptability. Specifically, with reference to Murray and Frenk [6], Levesque et al. [7], and Cu et al. [8], we clarified that the present study focuses in particular on spatial accessibility as geographic proximity. This information has been added to the revised manuscript (page 4, lines 62–74).

3-2. The introduction jumps from problem to methods without theoretical grounding

To present the theoretical foundation before the Methods section, we added a conceptual paragraph to the Introduction clarifying that this study treats spatial accessibility as the geographic prerequisite for potential access. We also explicitly distinguished it from availability, affordability, and acceptability [6–8]. This information has been added to the revised manuscript (page 4, lines 62–74).

3-3. Reference [10] (Shen & Tao) on spatial access is cited later but not introduced here

In response to the positioning of Shen and Tao, we moved the conceptual discussion of spatial accessibility to the Introduction so that it appears before the Methods section. In the final revised manuscript, we used Murray and Frenk [6], Levesque et al. [7], and Cu et al. [8] to present the multidimensional nature of access and to theoretically position the present study as an analysis limited to geographic proximity. This information has been added to the revised manuscript (page 4, lines 62–74).

3-4. RECOMMENDATION: Add a brief paragraph distinguishing spatial accessibility from other dimensions (affordability, acceptability, availability) and justify the focus on spatial proximity.

In accordance with your suggestion, we addressed the points raised in items 3-1 through 3-3 above.

4. Policy Context Underdeveloped

4-1. Line 44-45 mentions "critical policy concern" but doesn't elaborate

To clarify the specific meaning of “critical policy concern,” we added that, under population aging and population decline, whether residents can reach the healthcare services they need is a policy issue directly linked to the allocation of healthcare resources, the reduction of regional disparities, and the implementation of regional healthcare planning [2,3]. This information has been added to the revised manuscript (page 3, lines 43–45).

4-2. No mention of existing Japanese healthcare policies or planning frameworks

As part of the Japanese policy context, we explicitly added the Medical Care Act [4], the Ministry of Health, Labour and Welfare’s healthcare planning system [5], and the Tokyo Metropolitan Health and Medical Care Plan [20]. We also explained that secondary medical service areas serve as the practical unit for the division of hospital functions, the allocation of healthcare resources, and the evaluation of regional healthcare accessibility in a broad sense. This information has been added to the revised manuscript (page 3, line 50 to page 4, line 61; page 12, lines 181–182).

4-3. The later reference to secondary healthcare zones (line 75) appears without context

To ensure that secondary medical service areas do not appear abruptly in the Methods section, we added a paragraph on the policy context to the Introduction and introduced secondary medical service areas in advance as planning units based on the Medical Care Act [4] and the healthcare planning system [5]. This information has been added to the revised manuscript (page 3, line 50 to page 4, line 61).

4-4. RECOMMENDATION: Briefly introduce Japan's healthcare system structure and relevant policies driving interest in accessibility analysis.

By clarifying points 4-1 through 4-4 above, we were able to further clarify the conceptual framework of the study.

5. MATERIALS AND METHODS (Lines 73-155)

5-1. STUDY DESIGN SECTION MISSING (Lines 73-74)

The methods jump directly into "Study region and selected healthcare facilities" without describing the overall study design.

At the beginning of the Materials and Methods section, we newly added a “Study design” paragraph to clarify the overall study design before describing the study area and datasets. This information has been added to the revised manuscript (page 8, line 152 to page 9, line 165).

6. RECOMMENDATION: Add a brief study design paragraph covering:

In the revised manuscript, we explicitly stated that the present study is a cross-sectional spatial analysis. This information has been added to the revised manuscript (page 9, lines 153–154).

6-2. Timeframe

We specified the time points of the data used in the analysis. Specifically, we clarified that the study used data from the 2020 Population Census, 2021 data on elderly welfare facilities, 2022 public transportation data, and the December 2025 version of the MHLW Medical Information Network open data, and positioned the study as a cross-sectional spatial analysis combining data from different years. This information has been added to the revised manuscript (page 17, lines 242–249).

6-3. Overall analytical approach

We created Fig. 2 to present the overall workflow and Fig. 3 to illustrate the area-weighting method used for population estimation. In this way, we added a concise description of the overall analytical procedure, including the creation of fixed-distance buffers, estimation of population coverage, and evaluation of coverage for transportation-related facilities. This information has been added to the revised manuscript (Fig. 2; Fig. 3; page 9, lines 159–163; page 18, line 259 to page 21, line 323).

6-4. Software and tools used (ArcGIS Pro is mentioned line 126 but should be introduced earlier)

We described ArcGIS Pro earlier in the “Study design” subsection as the primary software used for geospatial processing and analysis. This information has been added to the revised manuscript (page 9, lines 163–165; page 20, lines 296–298).

7. FACILITY SELECTION RATIONALE INCOMPLETE (Lines 88-101)

7-1. Line 88: "The target facilities were of four types..." but no justification WHY these four

We explicitly stated in the manuscript that the target facilities were selected to represent the major components of Japan’s community-based healthcare delivery system, namely inpatient care (hospitals), outpatient care (clinics), dental care (dental clinics), pharmaceutical services (pharmacies), and elderly care–related resources (elderly welfare–related facilities). This information has been added to the revised manuscript (page 5, lines 81–83; page 13, lines 201–216).

7-2. Why exclude dental clinics after extracting them (line 99)?

We formally incorporated dental clinics, which had been excluded from the original manuscript, into the population coverage analysis and the Results section. As a result, the manuscript has been revised to provide a consistent comparison across five facility types: pharmacies, hospitals, clinics, dental clinics, and elderly welfare–related facilities. The inclusion of dental clinics as a study target has been stated in the revised manuscript (page 5, lines 81–83).

7-3. No operational definitions: What constitutes a "clinic" vs. "hospital"? Are these based on bed count, services, or legal designation?

Regarding the definitions of hospitals and clinics, we explicitly stated in the manuscript the standard Japanese classification based on the Medical Care Act [4]. Specifically, hospitals were defined as medical institutions with 20 or more beds, while clinics were defined as medical institutions with 19 or fewer beds or no beds (excluding facilities that provide dental services only). This information has been added to the revised manuscript (page 13, lines 204–208).

7-4. Elderly welfare facilities are defined collectively (lines 89-92) but the heterogeneity of these facilities is problematic for accessibility analysis

Regarding elderly welfare–related facilities, we clarified in the manuscript, with reference to the Elderly Welfare Act [21], that this category includes facility types with different internal functions. At the same time, we retained this category because of its policy importance in an aging society, and explicitly stated in the Methods section that the results should be interpreted as an aggregate indicator of older-adult care–related resources. This information has been added to the revised manuscript (page 13, lines 208–216).

7-5. RECOMMENDATION:

7-5-1. Provide clear inclusion/exclusion criteria for each facility type

We clarified these points in items 7-1 through 7-4 above.

7-5-2. Define facilities according to Japanese healthcare classifications

We clarified these points in items 7-1 through 7-4 above.

7-5-3. Justify the exclusion of dental clinics

In accordance with your suggestion, we incorporated dental clinics into the scope of the analysis. The inclusion of dental clinics as a study target has been stated in the revised manuscript (page 13, lines 201–204).

7-5-4. Consider analysing elderly welfare facilities separately or excluding them

This point was clarified in item 7-4 above and incorporated into the analysis.

8. DATA QUALITY AND VALIDATION (Lines 93-101)

8-1. No discussion of data completeness or accuracy.

We newly established a subsection titled “Data sources, data quality, and geocoding,” in which we summarized the sources of the facility data, the total numbers of facilities, the numbers of cases with missing coordinates, the methods used for coordinate completion, and the handling of data from different years.

This information has been added to the revised manuscript (page 16, line 220 to page 17, line 249).

8-2. Geocoding quality not assessed-what was the match rate from CSV Address Matching Service?

We specified the number and proportion of facilities that required coordinate completion. Among the target facilities after exclusion of the outlying islands, missing coordinates were identified for 24 pharmacies (0.36%), 1 hospital (0.17%), 72 clinics (0.59%), and 35 dental clinics (0.42%). We also clarified that these missing coordinates were supplemented using the CSV Address Matching Service [23] provided by the Center for Spatial Information Science, The University of Tokyo. This information has been added to the revised manuscript (page 12, lines 181–187; page 12, lines 226–231).

8-3. Different data vintages (2020 census, 2022 transportation, 2024 pharmacy) not acknowledged

We specified the time points

---

## [Decision Letter · Decision Letter 1]

11 May 2026

Spatial Analysis of Accessibility to Healthcare-Related Facilities in Tokyo Metropolis Using Geographic Information Systems

PONE-D-25-58835R1

Dear Dr. Nakamura,

We’re pleased to inform you that your manuscript has been judged scientifically suitable for publication and will be formally accepted for publication once it meets all outstanding technical requirements.

Kind regards,

Lingye Yao, Ph.D.

Academic Editor

PLOS One

Additional Editor Comments (optional):

Please ensure that the comments under point 6 and point 7 from Reviewer #2 are fully addressed during the proofreading stage.

Reviewers' comments:

Reviewer's Responses to Questions

**Comments to the Author**

1. If the authors have adequately addressed your comments raised in a previous round of review and you feel that this manuscript is now acceptable for publication, you may indicate that here to bypass the “Comments to the Author” section, enter your conflict of interest statement in the “Confidential to Editor” section, and submit your "Accept" recommendation.

Reviewer #1: All comments have been addressed

Reviewer #2: All comments have been addressed

2. Is the manuscript technically sound, and do the data support the conclusions?

Reviewer #1: Yes

Reviewer #2: Yes

3. Has the statistical analysis been performed appropriately and rigorously? 

Reviewer #1: Yes

Reviewer #2: Yes

4. Have the authors made all data underlying the findings in their manuscript fully available?

Reviewer #1: Yes

Reviewer #2: Yes

5. Is the manuscript presented in an intelligible fashion and written in standard English?

Reviewer #1: Yes

Reviewer #2: Yes

6. Review Comments to the Author

Reviewer #1: Manuscript ID: PONE-D-25-58835

Title: Spatial Analysis of Accessibility to Healthcare-Related Facilities in Tokyo Metropolis Using Geographic Information Systems

I thank the authors for their thorough response. The revised manuscript is substantially stronger, and the point-by-point reply, with precise page and line references, made re-evaluation straightforward.

The Introduction now includes the international GIS literature (Owen et al.; Luo and Wang; Luo and Qi), an additional Japanese study (Nakamura et al.), a clear conceptual framing of spatial accessibility as one dimension among several, three explicit hypotheses, and the policy context of Japan's secondary medical areas. The rationale for adopting a buffer-based approach is now stated transparently.

The Methods have been restructured into clearly labelled subsections, and the authors have in several instances gone beyond what I requested: dental clinics have been incorporated into the analysis rather than excluded; different buffer distances (400–3200 m for facilities, 250–3000 m for transport) are now used, addressing the differing walking tolerances by destination type; geocoding match rates and coordinate-correction procedures are reported explicitly; and operational definitions based on the Medical Care Act have been added. The renaming of "facility coverage rate" to "facility–transportation proximity rate" resolves the conceptual ambiguity I raised. Most importantly, a formal statistical analysis (Mann–Whitney U and Spearman's correlation) has been added, with results integrated into the Results section. The ethical statement, treatment of seasonal and operating-hour variability, and acknowledgement that no pilot testing was conducted have all been added.

On the urban–rural classification, the authors retained the distinction but complemented it with a continuous-variable analysis using population density and aging rate. This is one of the alternatives I explicitly suggested, and their justification, preserving alignment with the planning framework while capturing intra-metropolitan heterogeneity is convincing.

The revised Tables and Figures address my comments: Table 1 has been relabelled and expanded with % of Tokyo total and aging rate; the original Table 2 has been split appropriately; Figure 1 now includes a scale bar and an inset showing terrain and rivers; and the new Figure 3 provides both the explicit areal-weighting formula and a worked numerical example.

The Discussion has been meaningfully extended, integrating the supplementary statistical findings and offering a more nuanced reading of bus-stop proximity as a function of stop density rather than functional accessibility per se. The Limitations section now addresses the Euclidean-buffer assumption, mobility constraints, the absence of bicycle modelling, the uniform-population-distribution assumption, and temporal variability of public transport.

The authors have addressed essentially all my comments, and the manuscript is now substantially stronger in conceptual framing, methodological transparency, and statistical rigor.

I have no further substantive concerns and recommend acceptance.

Reviewer #2: The authors have addressed nearly all concerns; however, the response to Reviewer #2's point 6 appears to be truncated, and point 7 is missing. Nonetheless, the authors have accounted for these issues in the manuscript.

7. PLOS authors have the option to publish the peer review history of their article (what does this mean?). If published, this will include your full peer review and any attached files.

Reviewer #1: No

Reviewer #2: **Yes:** Kiki Adhinugraha

---

## [Editor Report · Acceptance letter]

PONE-D-25-58835R1

PLOS One

Dear Dr. Nakamura,

I'm pleased to inform you that your manuscript has been deemed suitable for publication in PLOS One. Congratulations! Your manuscript is now being handed over to our production team.

Kind regards,

on behalf of

Dr. Lingye Yao

Academic Editor

PLOS One